# Implicit Bias in Deep Linear Classification: Initialization Scale vs Training Accuracy

**Edward Moroshko**
edward.moroshko@gmail.com
Technion

**Blake Woodworth**
blake@ttic.edu
TTI Chicago

**Suriya Gunasekar**
suriya@ttic.edu
Microsoft Research

**Jason D. Lee**
jasonlee@princeton.edu
Princeton University

**Nathan Srebro**
nati@ttic.edu
TTI Chicago

**Daniel Soudry**
daniel.soudry@gmail.com
Technion

## Abstract

We provide a detailed asymptotic study of gradient flow trajectories and their implicit optimization bias when minimizing the exponential loss over "diagonal linear networks". This is the simplest model displaying a transition between "kernel" and non-kernel ("rich" or "active") regimes. We show how the transition is controlled by the relationship between the initialization scale and how accurately we minimize the training loss. Our results indicate that some limit behaviors of gradient descent only kick in at ridiculous training accuracies (well beyond $10^{-100}$). Moreover, the implicit bias at reasonable initialization scales and training accuracies is more complex and not captured by these limits.

## 1 Introduction

The optimization trajectory, and in particular the "implicit bias" determining which predictor the optimization algorithm leads to, plays a crucial role in learning with massively under-determined models, including deep networks, where many zero-error predictors are possible [*e.g.,* 22, 23, 29, 31]. Indeed, in several models we now understand how rich and natural implicit bias, often inducing sparsity of some form, can arise when training a multi-layer network with gradient descent [3, 11, 13, 16, 18–20, 29]. This includes low $\ell_1$ norm [30], sparsity in the frequency domain [13], low nuclear norm [11, 18], low rank [3, 25], and low higher-order total variations [6]. A different line of works focuses on how, in a certain regime, the optimization trajectory of neural networks, and hence also the implicit bias, stays near the initialization and mimics that of a kernelized linear predictor (with the kernel given by the tangent kernel) [1, 2, 4, 5, 7–9, 14, 17, 32]. In such a "kernel regime" the implicit bias corresponds to minimizing the norm in some Reproducing Kernel Hilbert Space (RKHS), and cannot yield the rich sparsity-inducing[1] inductive biases discussed above, and is perhaps not as "adaptive" as one might hope. It is therefore important to understand when and how learning is in the kernel regime, what hyper-parameters (*e.g.,* initialization, width, etc.) control the transition out and away from the kernel regime, and how the implicit bias (and thus the inductive bias driving learning) changes in different regimes and as a function of different hyper-parameters.

Initial work identified the **width** as a relevant hyper-parameter, where the kernel regime is reached when the width grows towards infinity [2, 8, 9, 14, 17, 32]. But subsequent work by Chizat et al. [7] pointed to the **initialization scale** as the relevant hyper-parameter, showing that models of any

width enter the kernel regime as the scale of initialization tends towards infinity. Follow-up work by Woodworth et al. [30] studied this transition in detail for regression with "diagonal linear networks" (see Section 3), showing how it is controlled by the interactions of **width**, **scale of initialization** and also **depth**. They obtained an exact expression for the implicit bias in terms of these hyper-parameters, showing how it transitions from $\ell_2$ (kernel) implicit bias when the width or initialization go to infinity, to $\ell_1$ ("rich" or "active" regime) for infinitesimal initialization, showing how a width of $k$ has an equivalent effect to increasing initialization scale by a factor of $\sqrt{k}$. Therefore, studying the effect of initialization scale can also be understood as studying the effect of width.

In this paper, we explore the transition and implicit bias for classification (as opposed to regression) problems, and highlight the effect of another hyper-parameter: **training accuracy**. A core distinction is that in classification with an exponentially decaying loss (such as the cross-entropy/logistic loss), the "rich regime" (*e.g.,* $\ell_1$) implicit bias is attained with *any finite initialization*, and not only with infinitesimal initialization as in regression [19, 20], and the scale of initialization does *not* effect the asymptotic implicit bias [12, 15, 28, 29]. This is in seeming contradiction to the fact that even for classification problems, the kernel regime *is* attained when the scale of initialization grows [7]. How can one reconcile the implicit bias not depending on the scale of initialization, with the kernel regime (and thus RKHS implicit bias)

|  | $\alpha < \infty$ | $\alpha \to \infty$ |
|---|---|---|
| $\epsilon > 0$ |  | Kernel [7] |
| $\epsilon \to 0$ | Rich [19] | **This Work** |

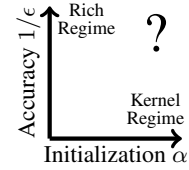

Figure 1: Kernel and rich limits

being reached as a function of initialization scale? The answer is discussed in detail in Section 2 and depicted in Figure 1: with the decaying losses used in classification, we can never get the loss to zero, only drive it arbitrarily close to zero. If we train infinitely long, then indeed we will eventually reach "rich" bias even for arbitrarily large initialization. On the other hand, if we only consider optimizing to within some arbitrarily high accuracy (*e.g.,* to loss $10^{-10}$), we will find ourselves in the kernel regime as the initialization scale grows. But how do these hyper-parameters interact? Where does this transition happen? How accurately do we need to optimize to exit the "kernel" regime? To get the "rich" implicit bias? And what happens in between? What happens if both initialization scale and training accuracy go to infinity? And how is this affected by the depth?

To answer these questions quantitatively, we consider a concrete model, and study classification using diagonal linear networks. This is arguably the simplest model that displays such transitions,[2] and as such has already been used to understand non-RKHS implicit bias and dynamics [10, 30].

We consider minimizing the exp-loss of a $D$-layer diagonal linear network on separable data, starting from initialization at scale $\alpha$ and optimizing until the training loss reaches a value of $\epsilon$. In this case, the two extreme regimes depicted in Figure 1 correspond to implicit biases given by the $\ell_2$ norm in the kernel regime and by the $\ell_{2/D}$ quasi-norm when $\epsilon \to 0$ for fixed $\alpha$. We consider $\alpha \to \infty$ and optimizing to within $\epsilon(\alpha) \to 0$ and ask what happens for different joint behaviours of $\alpha$ and $\epsilon$.

- We identify $\epsilon(\alpha) = \exp\big(-\Theta(\alpha^D)\big)$ as the boundary of the kernel regime. When $\epsilon(\alpha) = \exp\big(-o\left(\alpha^D\right)\big)$, the optimization trajectory follows that of a kernel machine, and the implicit bias is given by the $\ell_2$ norm. But when $\epsilon(\alpha) = \exp\big(-\Omega\left(\alpha^D\right)\big)$, the trajectory deviates from this kernel behaviour.
- Under additional condition (concerning stability of the support vectors), we characterize the behaviour at the transition regime, when $\epsilon(\alpha) = \exp\big(-\Theta(\alpha^D)\big)$, and show that at this scaling, the implicit bias is given by Woodworth et al.'s $Q^D$ regularizer, which interpolates between the $\ell_1$-norm and the $\ell_2$-norm. This indicates roughly the following optimization trajectory for large enough initialization scale and under the support vector stability condition: we will first pass through the minimum $Q^D_\infty = \ell_2$-norm predictor (or more accurately, max-margin w.r.t. the $\ell_2$-norm), then traverse the path of minimizers of $Q^D_\mu$, for $\mu \in (\infty, 0)$, until we reach the minimum $Q^D_0 = \ell_1$-norm predictor. We confirm such behaviour in simulations, as well as deviations from it when the condition does not hold.
- As suggested by our asymptotic theory, simulations in Section 5 show that even at moderate initialization scales, extreme training accuracy is needed in order to reach the asymptotic

behaviour where the implicit bias is well understood. Rather, we see that even in tiny problems, at moderate and reasonable initialization scales and training accuracies, the implicit bias behaves rather different from both the "kernel" and "rich" extremes, suggesting that further work is necessary to understand the effective implicit bias in realistic settings.

**Notation**   For vectors $\mathbf{z}, \mathbf{v}$, we denote by $\mathbf{z}^k$, $\exp(\mathbf{z})$, and $\log(\mathbf{z})$ the element-wise $k$th power, exponential, and natural logarithm, respectively; $\mathbf{z} \circ \mathbf{v}$ denotes element-wise multiplication; and $\mathbf{z} \propto \mathbf{v}$ implies that $\mathbf{z} = \gamma \mathbf{v}$ for a positive scalar $\gamma$. $\mathbf{1}$ denotes the vector of all ones.

## 2   Kernel and rich regimes in classification

We consider models as mappings $f : \mathbb{R}^p \times \mathcal{X} \to \mathbb{R}$ from trainable parameters $\mathbf{u} \in \mathbb{R}^p$ and input $\mathbf{x} \in \mathcal{X}$ to predictions. We denote by $F(\mathbf{u}) : \mathbf{x} \mapsto f(\mathbf{u}, \mathbf{x})$ the function implemented by the parameters $\mathbf{u}$. We will focus on models that are $D$-homogeneous in $\mathbf{u}$ for some $D > 0$, *i.e.,* such that $\forall c > 0, F(c\mathbf{u}) = c^D F(\mathbf{u})$. This includes depth-$D$ linear and ReLU networks.

We consider minimizing $\mathcal{L}(\mathbf{u}) = \frac{1}{N} \sum_{n=1}^N \ell(f(\mathbf{u}, \mathbf{x}_n), y_n)$ for a given  dataset $\{(\mathbf{x}_n, y_n) : n = 1, 2, \ldots N\}$ where $\ell : \mathbb{R} \times \mathcal{Y} \to \mathbb{R}$ is a loss function. We will be mostly focus on binary classification problems, where $y_n \in \{-1, 1\}$, and with the exp-loss $\ell(\hat{y}, y) = \exp(-\hat{y}y)$, which has the same tail behaviour and thus similar asymptotic properties as the logistic or cross-entropy loss [*e.g.,* 19, 28, 29]. All our results and discussion refer to the exp-loss unless explicitly stated otherwise. We are concerned with understanding the trajectory of gradient descent, which we consider at the limit of infintesimal stepsize, yielding the gradient flow dynamics,

$$\dot{\mathbf{u}}(t) = -\nabla \mathcal{L}(\mathbf{u}(t)) , \tag{1}$$

where here and throughout $\dot{\mathbf{u}} = \frac{\mathrm{d}\mathbf{u}}{\mathrm{d}t}$.

Along the gradient flow path, $\mathcal{L}(\mathbf{u}(t))$ is monotonically decreasing, and we consider cases where the loss is indeed minimized, *i.e.,* converges to 0 for $t \to \infty$. However, if we stop the optimization trajectory at a large but finite $t$, which is what we do in practice, we optimize to some positive training loss $\epsilon(t) = \mathcal{L}(\mathbf{u}(t))$. We define $\tilde{\gamma}(t) = -\log(\epsilon(t))$ as the *training accuracy*. $\tilde{\gamma}(t)$ can also be interpreted as the number of digits of the precision representing the training loss. This is related to the prediction margin $\gamma(t) = \min_n y_n \mathbf{x}_n^\top \mathbf{w}(t)$ as $\gamma(t) \le \tilde{\gamma}(t) \le \gamma(t) + \log(N)$ and was introduced as *smoothed margin* in Lyu and Li [19].

For classification problems, we consider **separable data**, *i.e.,* $\exists \mathbf{u}_* \in \mathbb{R}^p : \forall_n y_n f(\mathbf{u}_*, \mathbf{x}_n) > 0$, and so $\mathcal{L}(\gamma \mathbf{u}_*) \overset{\gamma \to \infty}{\longrightarrow} 0$. But especially in high dimensions, there are many such separating predictors. If $\mathcal{L}(\mathbf{u}(t)) \to 0$, which of these does $\mathbf{u}(t)$ converge to? Of course $\mathbf{u}(t)$ does not converge, since to approach zero error it must *diverge*. Therefore, instead of the limit of the parameters, we study the limit of the *decision boundary* of the resulting classifier, which is given by $F\left(\frac{\mathbf{u}(t)}{\|\mathbf{u}(t)\|}\right)$.

**Kernel Regime**   When the gradients $\nabla_\mathbf{u} f(\mathbf{u}, \mathbf{x}_n)$ do not change much during optimization, then $\mathbf{u}(t)$ behaves as if optimizing over a linearized model $\mathbf{u}$: $\bar{f}(\mathbf{u}, \mathbf{x}) = f(\mathbf{u}(0), \mathbf{x}) + \langle \mathbf{u} - \mathbf{u}(0), \phi(\mathbf{x}) \rangle$ where $\phi(\mathbf{x}) = \nabla_\mathbf{u} f(\mathbf{u}(0), \mathbf{x})$ is the feature map corresponding to the Tangent Kernel at initialization $K(\mathbf{x}, \mathbf{x}') = \langle \phi(\mathbf{x}), \phi(\mathbf{x}') \rangle$ [2, 8, 14]. Consider the trajectory $\bar{\mathbf{u}}(t)$ of gradient flow $\dot{\bar{\mathbf{u}}}(t) = -\nabla \bar{\mathcal{L}}(\bar{\mathbf{u}}(t))$ on the loss of this linearized model $\bar{\mathcal{L}}(\mathbf{u}) = \frac{1}{N} \sum_{n=1}^N \ell(\bar{f}(\mathbf{u}, \mathbf{x}_n), y_n)$. Chizat et al. [7] showed that any $D$-homogeneous model enters the kernel regime (*i.e.,* behaves like a linearized model) when the scale of the initialization is large:

**Theorem 1** (Adapted from Theorem 2.2 in Chizat et al. [7])**.** *For any fixed time horizon $T > 0$, and any $\mathbf{u}_0$ such that $F(\mathbf{u}_0) = 0$, and for the exp-loss, consider the two gradient flow trajectories $\mathbf{u}(t)$ and $\bar{\mathbf{u}}(t)$, respectively, both initialized with $\mathbf{u}(0) = \bar{\mathbf{u}}(0) = \alpha \mathbf{u}_0$, for $\alpha > 0$. Then $\lim_{\alpha \to \infty} \sup_{t \in [0,T]} \left\| F(\mathbf{u}(t)) - \bar{F}(\bar{\mathbf{u}}(t)) \right\| = 0$.*

For a linear model like $\bar{f}$, the gradient flow $\bar{\mathbf{u}}$ converges in direction to the maximum margin solution in the corresponding RKHS norm [28]. Combining this with Theorem 1, we have

$$\lim_{t \to \infty} \lim_{\alpha \to \infty} F\left(\frac{\mathbf{u}(t)}{\|\mathbf{u}(t)\|}\right) \propto \underset{f : \mathcal{X} \to \mathbb{R}}{\arg\min} \|f\|_K \text{ s.t. } \forall n : y_n f(\mathbf{x}_n) \ge 1, \tag{2}$$

where recall that $K$ is the Tangent Kernel at the initialization and $\|f\|_K$ is the RKHS norm with respect to this kernel. It is important to highlight the crucial difference here compared to the corresponding statement for the squared loss [7, Theorem 2.3]. For the squared loss we have that $\lim_{\alpha \to \infty} \sup_{t \in [0, \infty)} \left\| F(\mathbf{u}(t)) - \bar{F}(\bar{\mathbf{u}}(t)) \right\| = 0$, *i.e.*, the entire optimization trajectory converges uniformly to $\bar{\mathbf{u}}(t)$. But for the exp-loss, Theorem 1 only ensures convergence for prefixes of the path, up to finite time horizons $T$. The order of limits in (2) is thus crucial, while for the square loss the order of limits can be reversed.

**Rich regime** On the other hand, for any finite initialization $\alpha \mathbf{u}_0$, the limit direction of gradient flow, when optimized indefinitely, gives rise to a different limit solution [12, 19, 20]:

**Theorem 2** (Paraphrasing Theorem 4.4. in Lyu and Li [19]). *Assume that the gradient flow trajectory in* (1) *minimizes the loss, i.e.,* $\mathcal{L}(\mathbf{u}(t)) \to 0$. *Then, any limit point of* $\left\{ \frac{\mathbf{u}(t)}{\|\mathbf{u}(t)\|} : t > 0 \right\}$ *is along the direction of a KKT point of the following constrained optimization problem:*

$$\min_{\mathbf{u}} \|\mathbf{u}\|_2 \quad s.t. \ \forall n : y_n f(\mathbf{u}, \mathbf{x}_n) \geq 1. \tag{3}$$

Compared to the kernel regime in (2), Theorem 2 suggests[3] that

$$\lim_{\alpha \to \infty} \lim_{t \to \infty} F\left( \frac{\mathbf{u}(t)}{\|\mathbf{u}(t)\|} \right) \propto \text{ stationary points of } \min_{f:\mathcal{X} \to \mathbb{R}} \mathcal{R}(f) \quad \text{s.t. } \forall n : y_n f(\mathbf{x}_n) \geq 1,$$
$$\text{where } \mathcal{R}(f) = \min_{\mathbf{u}} \|\mathbf{u}\|_2 \ \text{s.t. } F(\mathbf{u}) = f. \tag{4}$$

To understand this double limit, note that Theorem 2 ensures convergence for every $\alpha$ separately, and so also as we take $\alpha \to \infty$. For neural networks including linear networks, $\mathcal{R}(f)$ captures rich and often sparsity inducing inductive biases (*e.g.,* nuclear norm, higher-order total variations, $\ell_p$ bridge penalty for $p \leq 1$) that are not captured by RKHS norms [11, 13, 15, 24, 26].

Contrasting (2) and (4) we see that if both the initialization scale $\alpha$ and the optimization time $t$ go to infinity, the order in which we take the limits is crucial in determining the implicit bias, matching the depiction in Figure 1. Roughly speaking, if $\alpha \to \infty$ first (*i.e.,* faster) and then $t \to \infty$, we end up in the kernel regime, but if $t \to \infty$ first (*i.e.,* faster), we can end up with rich implicit bias corresponding to $\mathcal{R}$. The main question we ask is: *where the transition from kernel to rich regime happens, and what is the implicit bias when $\alpha \to \infty$ and $t \to \infty$ together?*

Since the time $t$ for gradient flow does not directly correspond to actual "runtime", and is perhaps less directly meaningful, we instead consider the optimization trajectory in terms of the training loss $\epsilon(t) = \mathcal{L}(\mathbf{u}(t))$. For some loss tolerance parameter $\epsilon$, we follow the gradient flow trajectory until time $t$ such that $\mathcal{L}(t) = \epsilon$ and ask *what is the implicit bias when $\alpha \to \infty$ and $\epsilon \to 0$ together?*

## 3 Diagonal linear network of depth D

In the remainder of the paper we focus on depth-$D$ *diagonal linear networks*. This is a $D$-homogeneous model with parameters $\mathbf{u} = \begin{bmatrix} \mathbf{u}_+ \\ \mathbf{u}_- \end{bmatrix} \in \mathbb{R}^{2d}$ specified by:

$$f(\mathbf{u}, \mathbf{x}) = \langle \mathbf{u}_+^D - \mathbf{u}_-^D, \mathbf{x} \rangle \tag{5}$$

where recall that the exponentiation is element-wise.

As depicted in Figure 2, the model can be thought of as a depth-$D$ network, with $D - 1$ hidden linear layers (*i.e.,* the output is a weighted sum of the inputs), each consisting of $2d$ units, with the first hidden layer connected to the $d$ inputs and their negations (depicted in the figure as another fixed layer), each unit in subsequent hidden layers connected to only a single unit in the preceding hidden layer, and the single output unit connected to all units in the final hidden layer. That is, the weight

matrix at each layer $i = 1..D$ is a *diagonal* matrix $\mathrm{diag}(\mathbf{u}^i)$. This presentation has $2dD$ parameters, as every layer has a different weight matrix. However, it is easy to verify that if we initialize all layers to the same weight matrix, *i.e.,* with $\mathbf{u}^i = \mathbf{u}$, then the weight matrices will remain equal to each other throughout training, and so we can just use the $2d$ parameters in $\mathbf{u} \in \mathbb{R}^{2d}$ and take the weight matrix in every layer to be $\mathrm{diag}(\mathbf{u})$, recovering the model (5).

Since all operations in a linear neural net are linear, the model just implements a linear mapping from the input $\mathbf{x}$ to the output, and can therefor be viewed as an alternate parametrization of linear predictors. That is, the functions $F(\mathbf{u}) : \mathcal{X} \to \mathbb{R}$ implemented by the model is a linear predictor $F(\mathbf{u}) \in \mathcal{X}^*$ (since $\mathcal{X} = \mathbb{R}^d$, we also take $\mathcal{X}^* = \mathbb{R}^d$), and we can write $f(\mathbf{u}, \mathbf{x}) = \langle F(\mathbf{u}), \mathbf{x} \rangle$, where for diagonal linear nets $F(\mathbf{u}) = \mathbf{u}_+^D - \mathbf{u}_-^D$. In particular, for a trajectory $\mathbf{u}(t)$ in parameter space, we can also describe the corresponding trajectory $\mathbf{w}(t) = F(\mathbf{u}(t))$ of linear predictors.

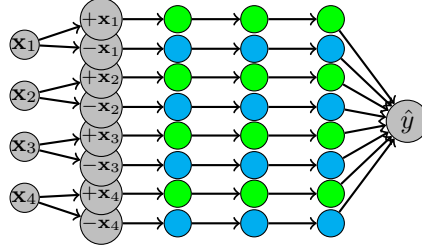

Figure 2: Diagonal linear network of depth 4.

The reason for using both the input features and their negation, and thus $2d$ units per layer, instead of a simpler model with $\mathbf{u} \in \mathbb{R}^d$ and $F(\mathbf{u}) = \mathbf{u}^D$ is two-fold: first, this allows the model to capture mixed sign linear predictors even with even depth. Second, this allows for scaling the parameters while still initializing at the zero predictor. In particular, we will consider initializing to $\mathbf{u}(0) = \alpha \mathbf{1}$, which for the diagonal neural net model (5) corresponds to $\mathbf{w}(0) = F(\mathbf{u}(0)) = 0$ regardless of the scale of $\alpha$. Such *unbiased initialization* was suggested by Chizat et al. [7] in order to avoid scaling a bias term when scaling the initialization.

Woodworth et al. [30] provided a detailed study of diagonal linear net regression using the square loss $\ell(\hat{y}, y) = (\hat{y} - y)^2$. They showed that for an underdetermined problem (*i.e.,* with multiple zero-error solutions), for any finite $\alpha$ the gradient flow trajectory with squared loss and initialization $\mathbf{u}(0) = \alpha \mathbf{1}$ converges to a zero-error (interpolating) solution minimizing the penalty $Q_{\alpha^D}^D$, where $Q_\mu^D$ is:

$$Q_\mu^D(\mathbf{w}) = \sum_{i=1}^d q^D\left(\frac{\mathbf{w}_i}{\mu}\right), \text{ where } q^D(z) = \begin{cases} 2 - \sqrt{4 + z^2} + z \cdot \mathrm{arcsinh}\left(\frac{z}{2}\right) & \text{for } D = 2 \\ \int_0^z h_D^{-1}(s)\, ds & \text{for } D > 2 \end{cases} \quad (6)$$

$$\text{and } h_D(s) = (1 - s)^{-\frac{D}{D-2}} - (1 + s)^{-\frac{D}{D-2}}.$$

For all $D \geq 2$, $Q_\mu^D$ with $\mu = \alpha^D$ interpolates between the $\ell_1$ norm as $\alpha \to 0$, which corresponds to the rich limit previously shown by Gunasekar et al. [11] and Arora et al. [3], and the $\ell_2$ norm as $\alpha \to \infty$, which is the RKHS norm defined by the tangent kernel at initialization.

Can we identify a similar transition behaviour between kernel and rich regimes with exponential loss?

## 4 Theoretical Analysis: Between the Kernel and Rich Regimes

We are now ready to state our results that describe the limit behaviour of gradient flow when the initialization scale $\alpha$ and the training accuracy $1/\epsilon$ go to infinity together for classification with linear diagonal networks and the exp-loss on separable data.

First, we establish that if the data is separable, even though the objective $\mathcal{L}(\mathbf{u})$ is non-convex, gradient flow will minimize it, *i.e.,* we will have $\mathcal{L}(\mathbf{u}(t)) \to 0$. We furthermore obtain a quantitative bound on how fast the training loss decreases as a function of the initialization scale $\alpha$, the depth $D$ and the $\ell_2$ separation margin of the data, $\gamma_2 = \max_{\|\mathbf{w}\|_2 = 1} \min_n y_n \mathbf{x}_n^\top \mathbf{w}$:

**Lemma 3.** *For $D \geq 2$, any fixed $\alpha$, and $\forall t$, $\mathcal{L}(\mathbf{u}(t)) \leq \frac{1}{1 + 2D^2 \alpha^{2D-2} \gamma_2^2 t}$.*

The proof appears in Appendix D.

We now turn to ask which separating classifier we would get to, if optimizing to within training accuracy $\epsilon$ and initialization of scale $\alpha$, in terms of the relationship between these two quantities. To capture this relationship, we consider a mapping $\epsilon(\alpha)$ such that $\epsilon : \mathbb{R}_{++} \to (0, 1]$ is strictly monotonic and $\lim_{\alpha \to \infty} \epsilon(\alpha) = 0$. We call $\epsilon(\alpha)$ the *stopping accuracy function*. For each $\alpha$, we follow the gradient flow trajectory until time $T_\alpha$ such that $\mathcal{L}(\mathbf{u}(T_\alpha)) = \epsilon(\alpha)$ and denote $\hat{\mathbf{w}}_\alpha = \frac{\mathbf{w}(T_\alpha)}{\gamma(T_\alpha)}$. We study the limit point $\hat{\mathbf{w}} = \lim_{\alpha \to \infty} \hat{\mathbf{w}}_\alpha$ for different $\epsilon(\alpha)$, assuming this limit exists.

**The Kernel Regime** We start with showing that if the stopping accuracy function $\epsilon(\alpha)$ goes to zero slowly enough, namely if $\log 1/\epsilon$ is sub-linear in $\alpha^D$, then with large initialization we obtain the $\ell_2$ bias of the kernel regime:

**Theorem 4.** *For $D \geq 2$, if $\epsilon(\alpha) = \exp\left(-o(\alpha^D)\right)$ then $\hat{\mathbf{w}} = \operatorname{argmin}_{\mathbf{w}} \|\mathbf{w}\|_2$ s.t. $\forall n, y_n \mathbf{w}^\top \mathbf{x}_n \geq 1$.*

The proof for $D = 2$ appears in Appendix F.1 and the proof for $D > 2$ appears in Appendix G.1.

**Escaping the kernel regime** Theorem 4 shows that escaping the kernel regime requires optimizing to higher accuracy, such that $\log 1/\epsilon$ is at least linear in the initialization scale $\alpha^D$. Complementing Theorem 4, we show a converse: that indeed the linear scaling $\log 1/\epsilon = \Theta(\alpha^D)$ is the transition point out of kernel regime, and once $\log 1/\epsilon = \Omega(\alpha^D)$ we no longer obtain the kernel $\ell_2$ bias.

To show this, we first identify a condition about the stability of support vectors[4] for a dataset $S$.

**Condition 5** (Stability condition). *A dataset $S = \{(\mathbf{x}_n, y_n) : n = 1, 2, \dots N\}$ and a stopping accuracy function $\epsilon(\alpha)$ satisfy the stability condition, if for all $k \in [N]$ such that $y_k \mathbf{x}_k^\top \hat{\mathbf{w}} > 1$, and large enough $\alpha$, there exists $\epsilon^\star(\alpha) = \exp\left(-o(\alpha^D)\right)$ and $\rho_0 > 1$ such that $\forall t$ with $\mathcal{L}(\mathbf{w}(t)) \in [\epsilon(\alpha), \epsilon^\star(\alpha)] : \frac{y_k \mathbf{x}_k^\top \mathbf{w}(t)}{\gamma(t)} \geq \rho_0$.*

*We further say that the stability condition holds uniformly for a given dataset $S$ if there exists $\rho_0$ such that the above condition holds for* all *stopping functions $\epsilon(\alpha) = \exp\left(-\Omega(\alpha^D)\right)$.*

If $\mathbf{w}(t)$ indeed follows the trajectory of the linearized model corresponding to the Neural Tangent Kernel, then Condition 5 holds (uniformly) for almost all datasets, see details in Appendix E. Therefore, if Condition 5 does not hold, it follows that we no longer follow the trajectory of this kernel model.

On the other hand, if Condition 5 does hold we show in the following Theorem 6 that when $\epsilon = \exp\left(-\Theta(\alpha^D)\right)$ we will be in the intermediate regime, leading to max-margin solution with respect to $Q^D$ function in (6), and again deviate from the kernel regime.

**Theorem 6.** *Under Condition 5, for $D \geq 2$, if $\lim_{\alpha \to \infty} \frac{\alpha^D}{\log(1/\epsilon(\alpha))} = \mu > 0$, then for $Q_\mu^D$ as defined in (6), $\hat{\mathbf{w}} = \operatorname{argmin}_{\mathbf{w}} Q_\mu^D(\mathbf{w})$ s.t. $\forall n, y_n \mathbf{x}_n^\top \mathbf{w} \geq 1$.*

The proof appears in Appendix F.2 (for $D = 2$) and Appendix G.2 (for $D > 2$). In the proof, we show that the following KKT conditions hold. The result then follows from convexity of $Q_\mu^D(\mathbf{w})$.

$$\exists \nu \in \mathbb{R}_{\geq 0}^N \text{ s.t.} \quad \nabla Q_\mu^D(\hat{\mathbf{w}}) = \sum_{n=1}^{N} \nu_n y_n \mathbf{x}_n, \quad \forall n : y_n \mathbf{x}_n^\top \hat{\mathbf{w}} \geq 1, \quad \forall n : \nu_n \left(y_n \mathbf{x}_n^\top \hat{\mathbf{w}} - 1\right) = 0.$$

**The Rich Limit** Above we saw that $\log 1/\epsilon$ being linear in $\alpha^D$ is enough to leave the kernel regime, *i.e.,* with this training accuracy and beyond the trajectory no longer behaves as if we were training a kernel machine. We also saw that under Condition 5, when $\log 1/\epsilon$ is exactly linear in $\alpha^D$, we are in a sense in an "transition" regime, with bias given by the $Q_\mu^D$ penalty which interpolates between $\ell_2$ (kernel) and $\ell_1$ (the rich limit for $D = 2$). Next we show that once the accuracy $\log 1/\epsilon$ is super-linear, and again under Condition 5, we are firmly at the rich limit:

**Theorem 7.** *Under Condition 5, for $D \geq 2$ if $\epsilon(\alpha) = \exp\left(-\omega(\alpha^D)\right)$ then $\hat{\mathbf{w}} = \operatorname{argmin}_{\mathbf{w}} \|\mathbf{w}\|_1$ s.t. $\forall n, y_n \mathbf{x}_n^\top \mathbf{w} \geq 1$.*

*For $D = 2$ the result holds also with a weaker condition, when $\epsilon^\star(\alpha)$ in Condition 5 is replaced with $\epsilon^\star(\alpha) = \exp\left[-o\left(\alpha^2 \log \frac{\log(1/\epsilon(\alpha))}{\alpha^2}\right)\right]$.*

The proof for $D = 2$ appears in Appendix F.3 and the proof for $D > 2$ appears in Appendix G.3.

For $D > 2$ we know from Theorem 2 that the implicit bias in the rich limit is given by an $\ell_{2/D}$ quasi norm penalty, and not by the $\ell_1$ penalty as in Theorem 7. It follows that when Condition 5 holds, the $\ell_1$ max-margin predictor must also be a first order stationary point of the $\ell_{2/D}$ max-margin problem. As we demonstrate in Section 5, this is certainly not always the case, and for many problems the $\ell_1$

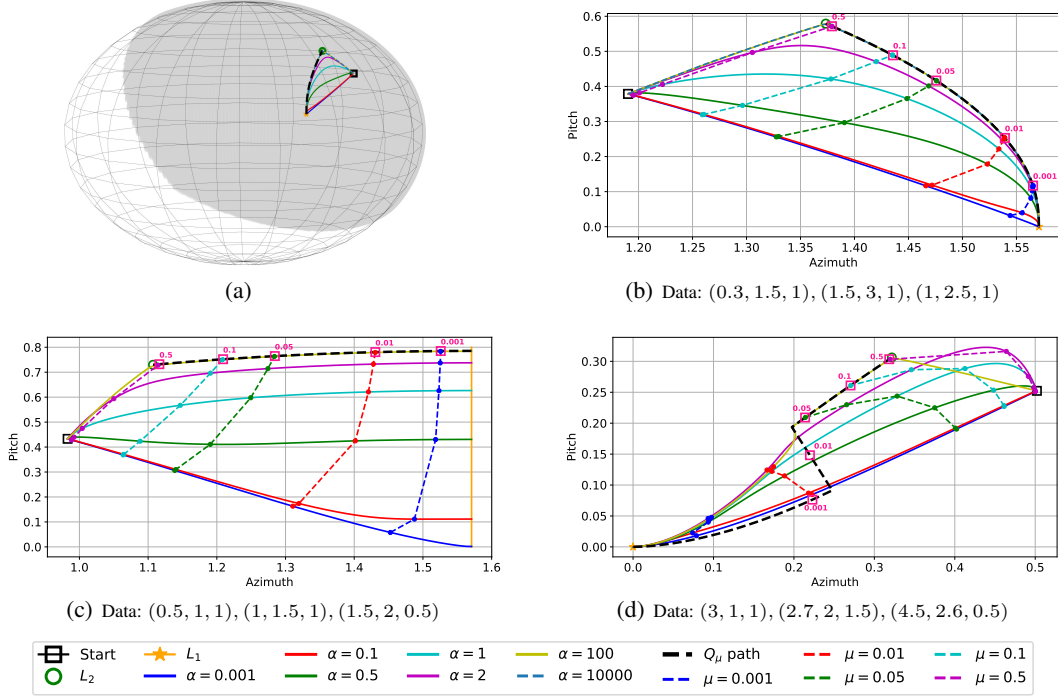

(a)

(b) Data: $(0.3, 1.5, 1), (1.5, 3, 1), (1, 2.5, 1)$

(c) Data: $(0.5, 1, 1), (1, 1.5, 1), (1.5, 2, 0.5)$

(d) Data: $(3, 1, 1), (2.7, 2, 1.5), (4.5, 2.6, 0.5)$

| Start | $L_1$ | $\alpha = 0.1$ | $\alpha = 1$ | $\alpha = 100$ | $Q_\mu$ path | $\mu = 0.01$ | $\mu = 0.1$ |
| $L_2$ | $\alpha = 0.001$ | $\alpha = 0.5$ | $\alpha = 2$ | $\alpha = 10000$ | $\mu = 0.001$ | $\mu = 0.05$ | $\mu = 0.5$ |

Figure 3: Optimization trajectories for 3 simple datasets in depth 2 linear diagonal network (b-d). Each point in Azimuth-Pitch plane represents a normalized classifier $^{\mathbf{w}}/\|\mathbf{w}\|_2$. The curves corresponding to $\alpha$ in the legend are the entire gradient flow trajectories initialized with the respective $\alpha$. The curves corresponding to $\mu$ are end points of gradient flow trajectories for different $\alpha$ with stopping criteria set as $\epsilon(\alpha) = \exp(-\alpha^2/\mu)$. The pink squares represent the directions along the $Q_\mu$ max-margin path for the appropriate $\mu$ marked near the square. The dynamics in (b) takes place on a small part of the sphere as shown in (a), where the grey area represents all separating directions.

max-margin is not a stationary point for the $\ell_{2/D}$ max-margin problem—in those cases Condition 5 does not hold. It might well be possible to show that a super-linear scaling is sufficient to reach the rich limit, be it the $\ell_1$ max-margin for depth two, or the $\ell_{2/D}$ max-margin (or a stationary point for this non-convex criteria) for higher depth, and we hope future work will address this issue.

**Role of depth** From Theorem 6 we have that asymptotically $\epsilon(\alpha) = \exp(-\alpha^D/\mu)$. Woodworth et al. [30] analyzed the $Q_\mu^D$ function[5] and concluded that in order to have $\delta$ approximation to $\ell_1$ limit (achieved for $\mu \to 0$) we need to have $\mu = \exp(-1/\delta)$ for $D = 2$, and $1/\mu = \mathrm{poly}(1/\delta)$ for $D > 2$. We conclude that in order to have $\delta$ approximation to $\ell_1$ limit we need the training accuracy to be $\epsilon = \exp(-\alpha^D \exp(1/\delta))$ for $D = 2$, and $\epsilon = \exp(-\alpha^D \mathrm{poly}(1/\delta))$ for $D > 2$. Thus, depth can mitigate the need to train to extreme accuracy. We confirm such behaviour in simulations.

## 5 Numerical Simulations and Discussion

We numerically study optimization trajectories to see whether we can observe the asymptotic phenomena studied at finite initialization and accuracy. We focus on low dimensional problems, where we can plot the trajectory in the space of predictors. In all our simulations we employ the Normalized GD algorithm, where the gradient is normalized by the loss itself, to accelerate convergence [21]. The learning rate was small enough to ensure gradient flow-like dynamics (always below $10^{-3}$).

**Gradient flow trajectories** In Figure 3 we plot trajectories for training depth $D = 2$ diagonal linear networks in dimension $d = 3$, on several constructed datasets, each consisting of three points.

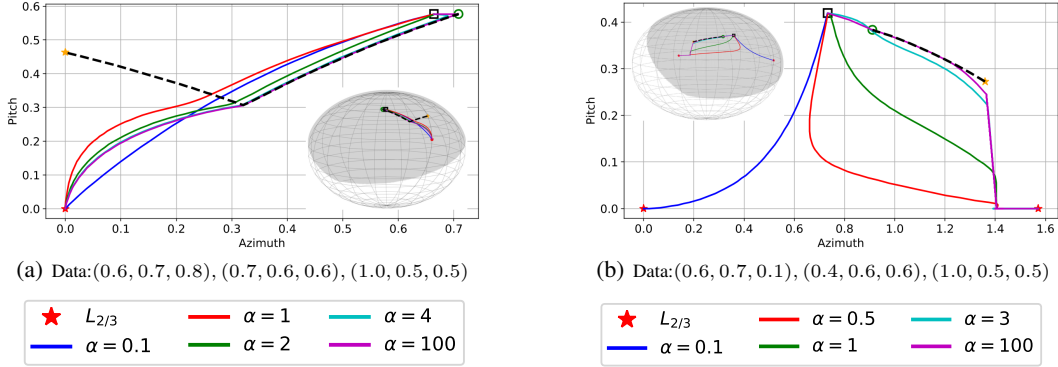

(a) Data: $(0.6, 0.7, 0.8), (0.7, 0.6, 0.6), (1.0, 0.5, 0.5)$      (b) Data: $(0.6, 0.7, 0.1), (0.4, 0.6, 0.6), (1.0, 0.5, 0.5)$

Figure 4: Optimization trajectories for $D = 3$. The legend for $\ell_1$, $\ell_2$, start and $Q_\mu$ path is the same as in Figure 3

The trajectory $\mathbf{w}(t)$ in this case is in $\mathbb{R}^3$, and so the corresponding binary predictor given by the normalization $\mathbf{w}(t)/\|\mathbf{w}(t)\|_2$ lies on the sphere (panel (a)). We zoom in on a small section of the sphere and plot the trajectory of $\mathbf{w}(t)/\|\mathbf{w}(t)\|_2$ — the axes correspond to coordinates on the sphere (given as azimuth and pitch). The first step taken by gradient flow will be to the predictor proportional to the average of the data, $\frac{1}{N}\sum_n y_n \mathbf{x}_n$, and we denote this as the "start" point. The grey area in the sphere represents classifiers separating the data (with zero misclassification error), and thus directions where the loss can be driven to zero. The question of "implicit bias" is which of these classifiers the trajectory will converge to. With infinitesimal (small) stepsizes, the trajectories always remain inside this area (*i.e.*, just finding a separating direction is easy), and in a sense, the entire optimization trajectory is driven by the implicit bias.

Panel 3(b) corresponds to a simple situation, with a unique $\ell_1$-max-margin solution, and where the support vectors for the $\ell_2$-max-margin and $\ell_1$-max-margin are the same (although the solutions are not the same!), and so the support vectors do not change throughout optimization and Condition 5 holds uniformly. For large initialization scales ($\alpha = 100$ and $\alpha = 10000$, which are indistinguishable here), the trajectory behaves as the asymptotic theory tells us: from the starting point (average of the data), we first go to the $\ell_2$-max-margin solution (green circle, and recall that this is also the $Q_\infty^2$-max-margin solution), and then follow the path of $Q_\mu^2$-max-margin predictors for $\mu$ going from $\infty$ to zero (this path is indicated by the dashed black lines in the plots), finally reaching the $\ell_1$-max-margin predictor (orange star, and this is also the $Q_0^2$-max-margin solution). For smaller initialization scales, we still always reach the same endpoint as $\epsilon \to 0$ (as assured by the theory), but instead of first visiting the $\ell_2$-max-margin solution and traversing the $Q_\mu^2$ path, we head more directly to the $\ell_1$-max-margin predictor. This can be thought of as the effect of initialization on the implicit bias and kernel regime transition: with small initialization we will never see the kernel regime, and go directly to the "rich" limit, but with large initialization we will initially remain in the kernel regime (heading to the $\ell_2$-max-margin), and then, only when the optimization becomes very accurate, escape it gradually.

To see the relative effect of scale and training accuracy, and following our theory, we plot for different values of $\mu$, the different points along trajectories with initialization $\alpha$ such that we fix the stopping criteria as $\epsilon(\alpha) = \exp(-\alpha^2/\mu)$ (dashed cross-lines in the plots). Our theory indicates that for any value of $\mu$, as $\alpha \to \infty$, the dashes line would converge to the $Q_\mu^2$-max-margin (a specific point on the $Q_\mu^2$ path), and this is indeed confirmed in Panel 3(b), where the dashed lines converge to pink squares, which correspond to points along the $Q_\mu^2$ path for the appropriate $\mu$ values. The clear correspondence between points with the same relationship $\mu$ between initialization and accuracy confirms that also at relatively small initialization scales, this parameter is the relevant relationship between them.

From the value $\mu$ we can also extract the actual training accuracy. E.g., we see that for a relatively large initialization scale $\alpha = 100$, escaping the kernel regime (getting to the first pink square just removed from the $\ell_2$-max-margin solution, with $\mu = 0.5$), requires optimizing to loss $\epsilon = \exp(-10^4/0.5) \approx 10^{-8700}$, while getting close to the asymptotic limit of the trajectory (the last pink square, with $\mu = 0.001$) requires $\epsilon \approx 10^{-4\cdot10^6}$ (that's four millions digits of precision). Even with reasonable initialization at scale $\alpha = 1$, getting to this limit requires $\epsilon \approx 10^{-434}$.

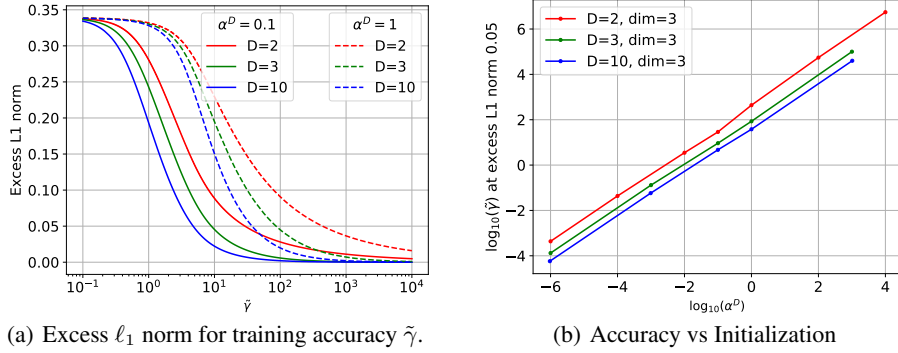

(a) Excess $\ell_1$ norm for training accuracy $\tilde{\gamma}$.

(b) Accuracy vs Initialization

Figure 5: In (a) we plot the excess $\ell_1$ norm, defined as $\|\mathbf{w}(t)\|_1 / \|\mathbf{w}_{\ell_1}\|_1 - 1$ where $\mathbf{w}_{\ell_1}$ is the $\ell_1$ max-margin (minimum norm) solution, as a function of $\tilde{\gamma}$. For a fixed excess $\ell_1$ norm of 0.05, in (b) we plot how long we need to optimize, given some initialization scale, to obtain the 5% closeness to $\ell_1$ max-margin solution.

Panel (c) displays the trajectories for another dataset where Condition 5 holds uniformly, but the $\ell_1$-max-margin solution is not unique. Although will always (eventually) converge to a $\ell_1$-max-margin solution, which one we converge to, and thus the implicit bias, *does* depends on the initialization. Panel (d) shows a situation where the support vectors for the $\ell_2$-max-margin and $\ell_1$-max-margin solutions are different, and so change during optimization, and Condition 5 does not hold uniformly. The condition does hold for $\mu > \mu_0$ where $\mu_0 \approx 0.04$, but does not hold otherwise. In this case, even for very large initialization scales $\alpha$, the trajectory does not follow the $Q_\mu^2$ path entirely. It is interesting to note that the $Q_\mu^2$ is not smooth, which perhaps makes it difficult to follow. Around the kink in the path (at $\mu = \mu_0$), it does seem that larger initialization scales are able to follow it a bit longer, and so perhaps with huge initialization (beyond our simulation ability), the trajectory *does* follow the $Q_\mu^2$ path. Whether the trajectory follows the path for sufficiently large initialization scales even when Condition 5 fails, or whether we can characterize its limit behaviour otherwise, remains an open question.

In Figure 4 we show optimization trajectories for depth 3 linear diagonal network where Condition 5 does not hold uniformly. As we can observe, in both cases the trajectory for large $\alpha$ will first go to $\ell_2$ max-margin, then stay near the $Q_\mu$ path, until the trajectory starts to deviate in a direction of $\ell_{2/3}$ max-margin solution, whereas the $Q_\mu$ path continues to the $\ell_1$ max-margin point. In Figure 4(b) we observe that there is a local minimum point at (0,0) and for small $\alpha$ the trajectory converges to it. Further discussion about convergence to local minima in high dimension appears in Appendix H.

**Initialization Scale vs Training Accuracy**    Using the same dataset from Figure 3(b), we examine the question: given some initialization scale, how long we need to optimize to be in the rich regime? Figure 5(a) demonstrates how the initialization and depth affect the convergence rate to the rich regime. Specifically, we chose two reasonable initialization scales $\alpha^D$, namely 0.1 and 1, and show their convergence to the $\ell_1$ max-margin solution as a function of the training accuracy $\tilde{\gamma}$ (recall that $\tilde{\gamma} = \log(1/\epsilon)$), for depths $D = 2, 3, 10$. We chose this dataset so that the rich regime is the same for all depths (*i.e.,* the minimum $\ell_1$ norm solution also corresponds to the minimum $\ell_{2/3}$ and $\ell_{2/10}$ quasi-norm solutions).

As previously discussed, even on such a small dataset (with three data-points) we are required to optimize to an incredibly high precision — in order to converge near the rich regime. Notably, the situation is improved and we converge faster to the rich regime when the initialization scale is smaller and/or the depth is larger. Unfortunately, taking the initialization scale to 0 will increase the time needed to escape the vicinity of the saddle point $\mathbf{u} = 0$. For example, Shamir [27] showed that exiting the initialization may take exponential-in-depth time.

In Figure 5(b) we examine the relative scale between (log) accuracy and (log) initialization needed to obtain 5% closeness to $\ell_1$ max-margin. Based on the asymptotic result in Theorem 6, we expect that $\tilde{\gamma} \propto \alpha^D$ or $\log(\tilde{\gamma}) = a \log(\alpha^D) + b$, for some constants $a = 1$ and $b \in \mathbb{R}$. And indeed, in Figure 5(b) we obtain $a = 1$, as expected. Note that although our theoretical results are valid for $\alpha \to \infty$, we obtain the same accuracy vs initialization rate also for small $\alpha$. Moreover, the intercept of the lines decreases when increasing the depth $D$, which matches the observed behaviour on Figure 5(a) and to the discussion about the effect of depth in Section 4.

## Broader Impact

The goal of this work is to shed light on the implicit bias hidden in the training process of deep networks. These results may enable a better understanding of how hyperparameters select the types of solutions that deep networks converge to, which in turn affect their final generalization performance and hidden biases. This could lead to better performance guarantees or to improved training algorithms which quickly converge to beneficial types of biases. Eventually, we believe progress on these fronts can transform deep learning from the current nascent "alchemy" age (where all the "knobs and levers" of the model and the training algorithm are tuned mostly heuristically during research and development), to a more mature field (like "chemistry"), which can be seamlessly integrated in many real world applications that require high performance, safety, and fair decisions.

Our guiding principal is that when studying a new or not-yet-understood phenomena, we should first study it in the simplest model that shows it, so as not to get distracted by possible confounders, and to enable a detailed analytic understanding, *e.g.,* when understanding or teaching many statistical issues, we would typically start with linear regression, understand the phenomena there, and *then* move on to more complex models. In the specific case here, one of the few models where we have an analytic handle on the implicit bias in the "rich" regime are linear diagonal networks, and it would be very optimistic to hope to get a detailed analytic description of the more complex phenomena we study in models where we can't even understand the endpoint.

## Acknowledgments and Disclosure of Funding

The research of DS was supported by the Israel Science Foundation (grant No. 31/1031), and by the Taub Foundation. This work was partially done while SG, JDL, NS, and DS were visiting the Simons Institute for the Theory of Computing. BW is supported by a Google Research PhD fellowship. JDL acknowledges support of the ARO under MURI Award W911NF-11-1-0303, the Sloan Research Fellowship, and NSF CCF 2002272.

## Footnotes

[1] Sparsity in an implicit space can also be understood as feature search or "adaptation". E.g. eigenvalue sparsity is equivalent to finding features which are linear combinations of input features, and sparsity in the space of ReLU units [*e.g.,* 26] corresponds to finding new non-linear features.

[2]Our guiding principal is that when studying a new or not-yet-understood phenomena, we should first carefully and fully understand it in the simplest model that shows it, so as not to get distracted by possible confounders, and to enable a detailed analytic understanding.

[3] Theorem 2 is suggestive of $\mathcal{R}(f)$ in (4) as the implicit induced bias in rich regime. However, although global minimizers of (3) and the RHS of (4) are equivalent, the same is not the case for stationary points. For the special cases of certain linear networks and the infinite width univariate ReLU network, stronger results for convergence in direction to the KKT points of (4) can be shown [6, 13, 16].

[4]Data-point $(\mathbf{x}_k, y_k)$ is a support vector at time $t$ if $y_k \mathbf{x}_k^\top \mathbf{w}(t) = \min_n y_n \mathbf{x}_n^\top \mathbf{w}(t) = \gamma(t)$.

[5]Note that $Q_\mu^D$ is defined for general $\mu$, and $\mu = \alpha^D$ is only for the square loss setting studied in [30]. Here $\mu$ depends both on $\alpha$ and $\epsilon$ (see Theorem 6).

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
