[Supplementary Material]

# Appendix

## A   Outline

This appendix is organized as follows: In Section B we provide preliminaries and notations used in the proofs. In Section C we prove auxiliary lemmas that characterize the dynamics of $\mathbf{w}(t)$ and bound the norm of $\mathbf{w}(t)$. In Section D we prove the loss bound in Lemma 3. In Section E we prove that Condition 5 holds for the linearized model. In the proofs we distinguish between the case $D = 2$ and $D > 2$ since the dynamics is different. In Section F we prove the results for $D = 2$ and in Section G we prove the results for $D > 2$. Finally, in Section H we provide additional simulation results and implementation details.

## B   Preliminaries and Notations for Proofs

To simplify notation in the proofs, without loss of generality we assume that $\forall n : y_n = 1$, as equivalently we can re-define $y_n \mathbf{x}_n$ as $\mathbf{x}_n$.

**Path parametrization:**   In the proofs we parameterize the optimization path in terms of $\tilde{\gamma}$. Recall that $\tilde{\gamma} = -\log \epsilon$ and $\tilde{\gamma}(t)$ is monotonically increasing along the gradient flow path starting from $\tilde{\gamma}(0) = -\log \epsilon(0) = 0$. Accordingly, the stopping criteria is $\tilde{\gamma}(\alpha) = \tilde{\gamma}(T_\alpha) = -\log \epsilon(\alpha)$. We also overload notation and denote $\mathbf{w}(\tilde{\gamma}') = \mathbf{w}(t_{\tilde{\gamma}'})$ and $\gamma(\tilde{\gamma}') = \gamma(t_{\tilde{\gamma}'})$ where $t_{\tilde{\gamma}'}$ is the unique $t$ such that $\tilde{\gamma}(t) = \tilde{\gamma}'$. Moreover, in this appendix we restate the conditions and theorems in terms of $\tilde{\gamma}$ rather than $\epsilon$.

**Notation:**   We use the following notations:

- $X = [\mathbf{x}_1, \ldots, \mathbf{x}_N] \in \mathbb{R}^{d \times N}$ denotes the data matrix.

- $\tilde{X} = [\tilde{\mathbf{x}}_1, \ldots, \tilde{\mathbf{x}}_N] \in \mathbb{R}^{2d \times N}$ denotes the augmented data matrix where $\tilde{\mathbf{x}}_n = \begin{bmatrix} \mathbf{x}_n \\ -\mathbf{x}_n \end{bmatrix} \in \mathbb{R}^{2d}$.

- $\bar{x}_i = \sum_{n=1}^N |x_{n,i}|$ where $x_{n,i}$ is the coordinate $i$ of $\mathbf{x}_n$. Also $\bar{x} = \max_i (\bar{x}_i)$.

- $x_{\max} = \max_n \|\mathbf{x}_n\|_2$.

- For some vector $\mathbf{z}$ we denote by $\mathrm{diag}(\mathbf{z})$ the diagonal matrix with diagonal $\mathbf{z}$, and $[\mathbf{z}]_i$ is the $i$ coordinate.

- The $\ell_2$ margin at time $t$ is $\gamma_2(t) = \frac{\min_n (\mathbf{x}_n^\top \mathbf{w}(t))}{\|\mathbf{w}(t)\|_2}$. Recall that $\gamma_2 = \max_{\|\mathbf{w}\|_2=1} \min_n \mathbf{x}_n^\top \mathbf{w}$.

- $\partial^\circ$ denotes the local sub-differential (Clarle's sub-differential) operator defined as
$$\partial^\circ h(\mathbf{z}) = \mathrm{conv}\left\{ \mathbf{v} : \exists \mathbf{z}_k \ \text{s.t.} \ \mathbf{z}_k \to \mathbf{z} \ \text{and} \ \nabla h(\mathbf{z}_k) \to \mathbf{v} \right\}.$$
Specifically, for $h(\mathbf{z}) = \|\mathbf{z}\|_1$:
$$\partial^\circ \|\mathbf{z}\|_1 = \left\{ \mathbf{v} \in \mathbb{R}^d : \forall i = 1, ..., d : \ -1 \leq v_i \leq 1 \ \text{and} \ z_i \neq 0 \Rightarrow v_i = \mathrm{sign}(z_i) \right\}.$$

- We denote $\mathbf{r}(t) = \frac{1}{N} \exp\left(-X^\top \mathbf{w}(t)\right)$. Note that $\|\mathbf{r}(t)\|_1 = \mathcal{L}(t) = \exp(-\tilde{\gamma}(t))$.

- We denote $A(t) = \mathrm{diag}\left(4\sqrt{\mathbf{w}^2(t) + 4\alpha^4 \mathbf{1}}\right)$. This matrix is used in the proofs for $D = 2$.

- For $D > 2$ let:
$$h_D(z) = (1 - z)^{-\frac{D}{D-2}} - (1 + z)^{-\frac{D}{D-2}} \quad , z \in (-1, 1). \tag{7}$$

Note that $h_D(z)$ is monotonically increasing, where $h_D(z) \overset{z \to -1}{\to} -\infty$ and $h_D(z) \overset{z \to 1}{\to} \infty$, and thus the inverse $h_D^{-1}$ is well defined, $h_D^{-1} : (-\infty, \infty) \to (-1, 1)$. In addition, it is easy to verify that for $z \in (-1, 1)$
$$h_D'(z) \doteq \frac{dh_D(z)}{dz} \geq \frac{2D}{D-2} \tag{8}$$
and
$$\lim_{z \to 0} \frac{h_D(z)}{z} = \frac{2D}{D-2}. \tag{9}$$

- We denote $A_D(t) = \mathrm{diag}\left(\alpha^{2D-2} D(D-2) h'_D \left(h_D^{-1}\left(\frac{\mathbf{w}(t)}{\alpha^D}\right)\right)\right)$. This matrix is used in the proofs for $D > 2$.

**Useful inequalities:** From the definitions of $\mathcal{L}(t)$ and $\tilde{\gamma}(t)$ we have that

$$\mathcal{L}(t) = \frac{1}{N} \sum_{n=1}^{N} \exp\left(-\mathbf{x}_n^\top \mathbf{w}(t)\right) = \exp\left(-\tilde{\gamma}(t)\right)$$

and thus

$$\frac{1}{N} \exp\left(-\gamma(t)\right) \leq \frac{1}{N} \sum_{n=1}^{N} \exp\left(-\mathbf{x}_n^\top \mathbf{w}(t)\right) \leq \frac{1}{N} N \exp\left(-\gamma(t)\right)$$

$$\Rightarrow \frac{1}{N} \exp\left(-\gamma(t)\right) \leq \exp\left(-\tilde{\gamma}(t)\right) \leq \exp\left(-\gamma(t)\right)$$

$$\Rightarrow \gamma(t) \leq \tilde{\gamma}(t) \leq \gamma(t) + \log(N) . \tag{10}$$

From eq. (10) we have that $\lim_{t\to\infty} \frac{\tilde{\gamma}(t)}{\gamma(t)} = 1$ and thus

$$\lim_{\alpha\to\infty} \frac{\tilde{\gamma}(T_\alpha)}{\gamma(T_\alpha)} = 1 . \tag{11}$$

In addition, using $\mathbf{x}_n^\top \mathbf{w}(t) \leq x_{\max} \|\mathbf{w}(t)\|_2$ we derive a lower bound on $\|\mathbf{w}(t)\|_2$ as following:

$$\mathcal{L}(t) = \frac{1}{N} \sum_{n=1}^{N} \exp\left(-\mathbf{x}_n^\top \mathbf{w}(t)\right)$$

$$\geq \frac{1}{N} N \exp\left(-x_{\max} \|\mathbf{w}(t)\|_2\right)$$

$$= \exp\left(-x_{\max} \|\mathbf{w}(t)\|_2\right)$$

and thus

$$x_{\max} \|\mathbf{w}(t)\|_2 \geq \log \frac{1}{\mathcal{L}(t)} = \tilde{\gamma}(t)$$

$$\Rightarrow \|\mathbf{w}(t)\|_2 \geq \frac{\tilde{\gamma}(t)}{x_{\max}} . \tag{12}$$

**Conditions:** We restate Condition 5 in terms of $\tilde{\gamma}$, *i.e.*, we substitute $\epsilon = \exp(-\tilde{\gamma})$. We consider two cases:

**Condition 8.** *For all $k \in [N]$ such that $\mathbf{x}_k^\top \hat{\mathbf{w}} > 1$, and large enough $\alpha$, there exists $\tilde{\gamma}^\star(\alpha) = o\left(\alpha^D\right)$ and $\rho_0 > 1$ such that $\forall \tilde{\gamma} \in [\tilde{\gamma}^\star(\alpha), \tilde{\gamma}(\alpha)] : \frac{\mathbf{x}_k^\top \mathbf{w}(\tilde{\gamma})}{\gamma(\tilde{\gamma})} \geq \rho_0$.*

**Condition 9.** *For all $k \in [N]$ such that $\mathbf{x}_k^\top \hat{\mathbf{w}} > 1$, and large enough $\alpha$, there exists $\tilde{\gamma}^\star(\alpha) = o\left(\alpha^2 \log \frac{\tilde{\gamma}(\alpha)}{\alpha^2}\right)$ and $\rho_0 > 1$ such that $\forall \tilde{\gamma} \in [\tilde{\gamma}^\star(\alpha), \tilde{\gamma}(\alpha)] : \frac{\mathbf{x}_k^\top \mathbf{w}(\tilde{\gamma})}{\gamma(\tilde{\gamma})} \geq \rho_0$.*

We prove the intermediate regime for $D \geq 2$ and the rich regime for $D > 2$ under Condition 8. To prove the rich regime for $D = 2$ the weaker Condition 9 will suffice.

## C   Auxiliary lemmas

### C.1   The case $D = 2$

**Lemma 10.** *For $D = 2$ and all $t$,*

$$\mathbf{w}(t) = 2\alpha^2 \sinh\left(4X \int_0^t \mathbf{r}(s)ds\right) \tag{13}$$

*and*

$$\frac{d\mathbf{w}(t)}{dt} = \frac{4}{N}\sqrt{\mathbf{w}^2(t) + 4\alpha^4\mathbf{1}} \circ X \exp\left(-X^\top \mathbf{w}(t)\right) = A(t)X\mathbf{r}(t) \tag{14}$$

*where $A(t) = \mathrm{diag}\left(4\sqrt{\mathbf{w}^2(t) + 4\alpha^4\mathbf{1}}\right)$.*

*Proof.* The gradient flow dynamics in the parameters space is given by

$$\dot{\mathbf{u}}(t) = -\nabla_{\mathbf{u}}\mathcal{L}(\mathbf{u}(t)) = \frac{2}{N}\mathbf{u}(t) \circ \tilde{X}\exp\left(-\tilde{X}^\top \mathbf{u}^2(t)\right). \tag{15}$$

It is easy to verify that the solution to eq. (15) can be written as

$$\mathbf{u}(t) = \mathbf{u}(0) \circ \exp\left(\frac{2}{N}\tilde{X}\int_0^t \exp\left(-\tilde{X}^\top \mathbf{u}^2(s)\right)ds\right) = \alpha\exp\left(\frac{2}{N}\tilde{X}\int_0^t \exp\left(-\tilde{X}^\top \mathbf{u}^2(s)\right)ds\right). \tag{16}$$

From (16) and $\mathbf{w} = \mathbf{u}_+^2 - \mathbf{u}_-^2$ we get eq. (13). Taking the derivative of eq. (13) we have

$$\dot{\mathbf{w}}(t) = \frac{8}{N}\alpha^2 \cosh\left(4X\int_0^t \mathbf{r}(s)ds\right) \circ X\exp\left(-X^\top \mathbf{w}(t)\right). \tag{17}$$

By combining eqs. (13) and (17) we get

$$\dot{\mathbf{w}}(t) = \frac{8}{N}\alpha^2 \cosh\left(\mathrm{arcsinh}\left(\frac{\mathbf{w}(t)}{2\alpha^2}\right)\right) \circ X\exp\left(-X^\top \mathbf{w}(t)\right).$$

Since $\cosh\left(\mathrm{arcsinh}(x)\right) = \sqrt{x^2 + 1}$ we get eq. (14). $\qquad\square$

**Lemma 11.** *For $D = 2$ and all $t$,*

$$\|\mathbf{w}(t)\|_\infty \le 2\alpha^2 \sinh\left(\frac{\bar{x}}{2\gamma_2^2\alpha^2}\tilde{\gamma}(t)\right).$$

*Proof.* Note that

$$\frac{d\mathcal{L}(t)}{dt} = \left(\nabla_{\mathbf{w}}\mathcal{L}(t)\right)^\top \frac{d\mathbf{w}(t)}{dt} = -\left(X\mathbf{r}(t)\right)^\top A(t)X\mathbf{r}(t)$$

$$\Rightarrow \frac{d\tilde{\gamma}(t)}{dt} = -\frac{1}{\mathcal{L}(t)}\frac{d\mathcal{L}(t)}{dt} = \frac{\left(X\mathbf{r}(t)\right)^\top A(t)X\mathbf{r}(t)}{\|\mathbf{r}(t)\|_1}.$$

From $A_{i,i}(t) \ge 8\alpha^2$ we have

$$\frac{d\tilde{\gamma}(t)}{dt} \ge \frac{8\alpha^2\|X\mathbf{r}(t)\|_2^2}{\|\mathbf{r}(t)\|_1}. \tag{18}$$

From Lemma 2 of [21] we have that

$$\|X\mathbf{r}(t)\|_2 \ge \gamma_2\|\mathbf{r}(t)\|_1. \tag{19}$$

Combining eqs. (18) and (19) we get

$$\frac{d\tilde{\gamma}(t)}{dt} \ge 8\alpha^2\gamma_2^2\|\mathbf{r}(t)\|_1 = 8\alpha^2\gamma_2^2\exp\left(-\tilde{\gamma}(t)\right). \tag{20}$$

We employ the dynamics equation $\dot{\mathbf{w}}(t) = \frac{4}{N}\sqrt{\mathbf{w}^2(t) + 4\alpha^4\mathbf{1}} \circ X\exp\left(-X^\top\mathbf{w}(t)\right)$ and change variables $t \to \tilde{\gamma}(t)$. Using eq. (20) we get that

$$\left|\frac{dw_i(\tilde{\gamma})}{d\tilde{\gamma}}\right| = \left|\frac{dw_i(t)}{dt}\frac{dt}{d\tilde{\gamma}}\right| \le \left(\sqrt{w_i^2(\tilde{\gamma}) + 4\alpha^4}\right)\left|\left[X\exp\left(-X^\top\mathbf{w}(\tilde{\gamma})\right)\right]_i\right|\frac{1}{2N\alpha^2\gamma_2^2\exp\left(-\tilde{\gamma}\right)}.$$

Using $\exp\left(-\mathbf{x}_k^\top\mathbf{w}(\tilde{\gamma})\right) \le N\exp\left(-\tilde{\gamma}\right)$ which follows from eq. (10) we get

$$\left|\frac{dw_i(\tilde{\gamma})}{d\tilde{\gamma}}\right| \le \frac{\bar{x}_i}{2\alpha^2\gamma_2^2}\sqrt{w_i^2(\tilde{\gamma}) + 4\alpha^4}$$

and by the Grönwall's inequality we get the desired bound

$$|w_i(\tilde{\gamma})| \le 2\alpha^2\sinh\left(\frac{\bar{x}_i}{2\gamma_2^2\alpha^2}\tilde{\gamma}\right) \le 2\alpha^2\sinh\left(\frac{\bar{x}}{2\gamma_2^2\alpha^2}\tilde{\gamma}\right).$$

$\qquad\square$

## C.2 The case $D > 2$

**Lemma 12.** *For $D > 2$ and all t,*

$$\mathbf{w}(t) = \alpha^D h_D \left( \alpha^{D-2} D (D-2) X \int_0^t \mathbf{r}(s) \, ds \right)$$

*and*

$$\frac{d\mathbf{w}(t)}{dt} = A_D(t) X \mathbf{r}(t)$$

*where $A_D(t) = \text{diag}\left( \alpha^{2D-2} D (D-2) h_D' \left( h_D^{-1} \left( \frac{\mathbf{w}(t)}{\alpha^D} \right) \right) \right)$.*

*Proof.* The gradient flow dynamics in the parameters space is given by

$$\dot{\mathbf{u}}(t) = -\nabla_{\mathbf{u}} \mathcal{L}(\mathbf{u}(t)) = \frac{D}{N} \mathbf{u}^{D-1}(t) \circ \tilde{X} \exp\left( -\tilde{X}^\top \mathbf{u}^D(t) \right) . \tag{21}$$

It is easy to verify that the solution to eq. (21) is

$$\mathbf{u}(t) = \left( \mathbf{u}^{2-D}(0) - \frac{D(D-2)}{N} \tilde{X} \int_0^t \exp\left( -\tilde{X}^\top \mathbf{u}^D(s) \right) ds \right)^{-\frac{1}{D-2}}$$

$$= \left( \alpha^{2-D} \mathbf{1} - \frac{D(D-2)}{N} \tilde{X} \int_0^t \exp\left( -\tilde{X}^\top \mathbf{u}^D(s) \right) ds \right)^{-\frac{1}{D-2}}$$

$$= \alpha \left( \mathbf{1} - \frac{\alpha^{D-2} D(D-2)}{N} \tilde{X} \int_0^t \exp\left( -\tilde{X}^\top \mathbf{u}^D(s) \right) ds \right)^{-\frac{1}{D-2}} . \tag{22}$$

From eq. (22) and $\mathbf{w} = \mathbf{u}_+^D - \mathbf{u}_-^D$ we get

$$\mathbf{w}(t) = \alpha^D \left[ \left( \mathbf{1} - \alpha^{D-2} D(D-2) X \int_0^t \mathbf{r}(s) \, ds \right)^{-\frac{D}{D-2}} \right.$$

$$\left. - \left( \mathbf{1} + \alpha^{D-2} D(D-2) X \int_0^t \mathbf{r}(s) \, ds \right)^{-\frac{D}{D-2}} \right] . \tag{23}$$

As $u_i(t) \geq 0$ for all $i$ (because $u_i(0) = \alpha > 0$; the gradient flow dynamics are continuous; and $u_i(t) = 0 \Rightarrow \dot{u}_i(t) = 0$) we get from eq. (22) that

$$-1 \leq \frac{\alpha^{D-2} D(D-2)}{N} X \int_0^t \exp\left( -\tilde{X}^\top \mathbf{u}^D(s) \right) ds \leq 1 . \tag{24}$$

Therefore we can write eq. (23) as

$$\mathbf{w}(t) = \alpha^D h_D \left( \alpha^{D-2} D(D-2) X \int_0^t \mathbf{r}(s) \, ds \right) \tag{25}$$

$$\Rightarrow \alpha^{D-2} D(D-2) X \int_0^t \mathbf{r}(s) \, ds = h_D^{-1} \left( \frac{\mathbf{w}(t)}{\alpha^D} \right) .$$

Taking the derivative of eq. (25) we get

$$\dot{\mathbf{w}}(t) = \alpha^D h_D' \left( \alpha^{D-2} D(D-2) X \int_0^t \mathbf{r}(s) \, ds \right) \circ \left( \alpha^{D-2} D(D-2) X \mathbf{r}(t) \right)$$

$$= \alpha^{2D-2} D(D-2) h_D' \left( h_D^{-1} \left( \frac{\mathbf{w}(t)}{\alpha^D} \right) \right) \circ (X \mathbf{r}(t)) . \tag{26}$$

$\square$

**Lemma 13.** *For $D > 2$ and all t,*

$$\| \mathbf{w}(t) \|_\infty \leq \alpha^D h_D \left( \frac{(D-2)\bar{x}}{2D\gamma_2^2 \alpha^D} \tilde{\gamma}(t) \right) .$$

*Proof.* Note that

$$\frac{d\mathcal{L}(t)}{dt} = (\nabla_{\mathbf{w}}\mathcal{L}(t))^\top \frac{d\mathbf{w}(t)}{dt} = -(X\mathbf{r}(t))^\top A_D(t) X\mathbf{r}(t)$$

$$\Rightarrow \frac{d\tilde{\gamma}(t)}{dt} = -\frac{1}{\mathcal{L}(t)}\frac{d\mathcal{L}(t)}{dt} = \frac{(X\mathbf{r}(t))^\top A_D(t) X\mathbf{r}(t)}{\|\mathbf{r}(t)\|_1}. \tag{27}$$

From eq. (8) we get a lower bound on the entries of $A_D(t)$, $A_D(t) \geq 2D^2\alpha^{2D-2}$. Combining with eq. (27) we get

$$\frac{d\tilde{\gamma}(t)}{dt} \geq \frac{2D^2\alpha^{2D-2}\|X\mathbf{r}(t)\|_2^2}{\|\mathbf{r}(t)\|_1}. \tag{28}$$

From eqs. (28), (19) we get

$$\frac{d\tilde{\gamma}(t)}{dt} \geq 2D^2\alpha^{2D-2}\gamma_2^2\|\mathbf{r}(t)\|_1 = 2D^2\alpha^{2D-2}\gamma_2^2\exp\left(-\tilde{\gamma}(t)\right). \tag{29}$$

We employ the dynamics equation eq. (26) and change variables $t \to \tilde{\gamma}(t)$. Using eq. (29) we get that

$$\left|\frac{dw_i(\tilde{\gamma})}{d\tilde{\gamma}}\right| = \left|\frac{dw_i(t)}{dt}\frac{dt}{d\tilde{\gamma}}\right| \leq \alpha^{2D-2}D(D-2)h_D'\left(h_D^{-1}\left(\frac{w_i(\tilde{\gamma})}{\alpha^D}\right)\right)\frac{\left|\left[X\exp\left(-X^\top\mathbf{w}(\tilde{\gamma})\right)\right]_i\right|}{2ND^2\alpha^{2D-2}\gamma_2^2\exp\left(-\tilde{\gamma}\right)}.$$

Using $\exp\left(-\mathbf{x}_k^\top\mathbf{w}(\tilde{\gamma})\right) \leq N\exp\left(-\tilde{\gamma}\right)$ which follows from eq. (10) we get

$$\left|\frac{dw_i(\tilde{\gamma})}{d\tilde{\gamma}}\right| \leq \frac{(D-2)\bar{x}_i}{2D\gamma_2^2}h_D'\left(h_D^{-1}\left(\frac{w_i(\tilde{\gamma})}{\alpha^D}\right)\right)$$

and by the Grönwall's inequality we get the desired bound

$$|w_i(\tilde{\gamma})| \leq \alpha^D h_D\left(\frac{(D-2)\bar{x}_i}{2D\gamma_2^2\alpha^D}\tilde{\gamma}\right) \leq \alpha^D h_D\left(\frac{(D-2)\bar{x}}{2D\gamma_2^2\alpha^D}\tilde{\gamma}\right).$$

$\square$

# D   Proof of Lemma 3

We prove the loss bound for $D \geq 2$, any fixed $\alpha$, and $\forall t$:

$$\mathcal{L}(t) \leq \frac{1}{1 + 2D^2\alpha^{2D-2}\gamma_2^2 t}.$$

*Proof.* We employ the Grönwall's inequality. For $D = 2$ from eq. (20) we get

$$\tilde{\gamma}(t) \geq \log\left(1 + 8\alpha^2\gamma_2^2 t\right) \tag{30}$$

and thus

$$\mathcal{L}(t) \leq \frac{1}{1 + 8\alpha^2\gamma_2^2 t}. \tag{31}$$

For $D > 2$ from eq. (29) we get

$$\tilde{\gamma}(t) \geq \log\left(1 + 2D^2\alpha^{2D-2}\gamma_2^2 t\right) \tag{32}$$

and thus

$$\mathcal{L}(t) \leq \frac{1}{1 + 2D^2\alpha^{2D-2}\gamma_2^2 t}. \tag{33}$$

Note that by substituting $D = 2$ in eq. (33) we get eq. (31), so (33) is correct for $D \geq 2$.   $\square$

# E   Condition 5 holds for the linearized model

We show that Condition 8, which is equivalent to Condition 5, holds for the linearized model. The linearized model is

$$\bar{f}\left(\bar{\mathbf{u}}, \mathbf{x}\right) = f\left(\bar{\mathbf{u}}\left(0\right), \mathbf{x}\right) + \nabla_{\mathbf{u}}^{\top} f\left(\bar{\mathbf{u}}\left(0\right), \mathbf{x}\right)\left(\bar{\mathbf{u}} - \bar{\mathbf{u}}\left(0\right)\right) .$$

For the diagonal linear network $f\left(\mathbf{u}, \mathbf{x}\right) = \mathbf{x}^{\top}\left(\mathbf{u}_+^D - \mathbf{u}_-^D\right)$, where $\mathbf{u} = \begin{bmatrix} \mathbf{u}_+ \\ \mathbf{u}_- \end{bmatrix} \in \mathbb{R}^{2d}$. Let $\bar{\mathbf{u}} = \begin{bmatrix} \bar{\mathbf{u}}_+ \\ \bar{\mathbf{u}}_- \end{bmatrix} \in \mathbb{R}^{2d}$. We consider the initialization $\mathbf{u}\left(0\right) = \bar{\mathbf{u}}\left(0\right) = \alpha \mathbf{1}$, thus

$$f\left(\bar{\mathbf{u}}\left(0\right), \mathbf{x}\right) = 0$$

$$\nabla_{\mathbf{u}} f\left(\bar{\mathbf{u}}\left(0\right), \mathbf{x}\right) = D\alpha^{D-1}\begin{bmatrix} \mathbf{x} \\ -\mathbf{x} \end{bmatrix}$$

$$\nabla_{\mathbf{u}}^{\top} f\left(\bar{\mathbf{u}}\left(0\right), \mathbf{x}\right)\bar{\mathbf{u}}\left(0\right) = 0$$

and we get

$$\bar{f}\left(\bar{\mathbf{u}}, \mathbf{x}\right) = D\alpha^{D-1}\mathbf{x}^{\top}\left(\bar{\mathbf{u}}_+ - \bar{\mathbf{u}}_-\right) .$$

Let $\bar{\mathbf{w}} = D\alpha^{D-1}\left(\bar{\mathbf{u}}_+ - \bar{\mathbf{u}}_-\right)$. Then $\bar{f}\left(\bar{\mathbf{w}}, \mathbf{x}\right) = \bar{\mathbf{w}}^{\top}\mathbf{x}$. We consider gradient flow $\frac{d\bar{\mathbf{u}}(t)}{dt} = -\nabla\bar{\mathcal{L}}\left(\bar{\mathbf{u}}\left(t\right)\right)$ where

$$\bar{\mathcal{L}}\left(\bar{\mathbf{u}}\left(t\right)\right) = \frac{1}{N}\sum_{n=1}^{N}\exp\left(-\bar{f}\left(\bar{\mathbf{u}}\left(t\right), \mathbf{x}_n\right)\right) .$$

Thus

$$\frac{d\bar{\mathbf{u}}_+\left(t\right)}{dt} = \frac{1}{N}D\alpha^{D-1}\sum_{n=1}^{N}\exp\left(-D\alpha^{D-1}\mathbf{x}_n^{\top}\left(\bar{\mathbf{u}}_+ - \bar{\mathbf{u}}_-\right)\right)\mathbf{x}_n$$

$$\frac{d\bar{\mathbf{u}}_-\left(t\right)}{dt} = -\frac{1}{N}D\alpha^{D-1}\sum_{n=1}^{N}\exp\left(-D\alpha^{D-1}\mathbf{x}_n^{\top}\left(\bar{\mathbf{u}}_+ - \bar{\mathbf{u}}_-\right)\right)\mathbf{x}_n$$

and

$$\frac{d\bar{\mathbf{w}}\left(t\right)}{dt} = D\alpha^{D-1}\left(\frac{d\bar{\mathbf{u}}_+\left(t\right)}{dt} - \frac{d\bar{\mathbf{u}}_-\left(t\right)}{dt}\right)$$

$$= \frac{2}{N}D^2\alpha^{2D-2}\sum_{n=1}^{N}\exp\left(-\mathbf{x}_n^{\top}\bar{\mathbf{w}}\left(t\right)\right)\mathbf{x}_n . \tag{34}$$

It follows that

$$\frac{d\bar{\mathcal{L}}\left(t\right)}{dt} = \left(\nabla_{\bar{\mathbf{w}}}\bar{\mathcal{L}}\left(t\right)\right)^{\top}\frac{d\bar{\mathbf{w}}\left(t\right)}{dt}$$

$$= \left(-\frac{1}{N}\sum_{n=1}^{N}\exp\left(-\mathbf{x}_n^{\top}\bar{\mathbf{w}}\left(t\right)\right)\mathbf{x}_n\right)^{\top}\left(\frac{2}{N}D^2\alpha^{2D-2}\sum_{n=1}^{N}\exp\left(-\mathbf{x}_n^{\top}\bar{\mathbf{w}}\left(t\right)\right)\mathbf{x}_n\right)$$

$$= -2D^2\alpha^{2D-2}\left\|\frac{1}{N}\sum_{n=1}^{N}\exp\left(-\mathbf{x}_n^{\top}\bar{\mathbf{w}}\left(t\right)\right)\mathbf{x}_n\right\|_2^2 .$$

Let $\bar{\bar{\gamma}}\left(t\right) = \log\frac{1}{\bar{\mathcal{L}}(t)}$. Then

$$\frac{d\bar{\bar{\gamma}}\left(t\right)}{dt} = -\frac{1}{\bar{\mathcal{L}}\left(t\right)}\frac{d\bar{\mathcal{L}}\left(t\right)}{dt} = \frac{1}{\bar{\mathcal{L}}\left(t\right)}2D^2\alpha^{2D-2}\left\|\frac{1}{N}\sum_{n=1}^{N}\exp\left(-\mathbf{x}_n^{\top}\bar{\mathbf{w}}\left(t\right)\right)\mathbf{x}_n\right\|_2^2 . \tag{35}$$

From Lemma 2 of [21] we know that

$$\left\|\frac{1}{N}\sum_{n=1}^{N}\exp\left(-\mathbf{x}_n^{\top}\bar{\mathbf{w}}\left(t\right)\right)\mathbf{x}_n\right\|_2^2 \geq \gamma_2^2\left(\frac{1}{N}\sum_{n=1}^{N}\exp\left(-\mathbf{x}_n^{\top}\bar{\mathbf{w}}\left(t\right)\right)\right)^2 = \gamma_2^2\bar{\mathcal{L}}^2\left(t\right) . \tag{36}$$

Combining eqs. (35) and (36) we get

$$\frac{d\bar{\bar{\gamma}}(t)}{dt} \geq 2D^2\alpha^{2D-2}\gamma_2^2\bar{\mathcal{L}}(t) = 2D^2\alpha^{2D-2}\gamma_2^2\exp\left(-\bar{\bar{\gamma}}(t)\right). \tag{37}$$

In addition,

$$\left\|\frac{1}{N}\sum_{n=1}^{N}\exp\left(-\mathbf{x}_n^\top\bar{\mathbf{w}}(t)\right)\mathbf{x}_n\right\|_2 \leq \frac{1}{N}\sum_{n=1}^{N}\exp\left(-\mathbf{x}_n^\top\bar{\mathbf{w}}(t)\right)\|\mathbf{x}_n\|_2 \leq x_{\max}\bar{\mathcal{L}}(t). \tag{38}$$

Combining eqs. (35) and (38) we get

$$\frac{d\bar{\bar{\gamma}}(t)}{dt} \leq 2D^2\alpha^{2D-2}x_{\max}^2\bar{\mathcal{L}}(t) = 2D^2\alpha^{2D-2}x_{\max}^2\exp\left(-\bar{\bar{\gamma}}(t)\right)$$

and by the Grönwall's inequality we get

$$\bar{\bar{\gamma}}(t) \leq \log\left(1 + 2D^2\alpha^{2D-2}x_{\max}^2 t\right)$$

$$\Rightarrow t \geq \frac{\exp\left(\bar{\bar{\gamma}}\right) - 1}{2D^2\alpha^{2D-2}x_{\max}^2}. \tag{39}$$

The $\ell_2$ max-margin solution is $\mathbf{w}_{\ell_2} = \sum_{n \in S_2}\nu_n\mathbf{x}_n$ where $S_2$ denotes the set of support vectors of $\mathbf{w}_{\ell_2}$. Let $\tilde{\mathbf{w}}$ be a vector that satisfies $\exp\left(-\mathbf{x}_n^\top\tilde{\mathbf{w}}\right) = \nu_n$ for $n \in S_2$. Such $\tilde{\mathbf{w}}$ exists for almost all datasets, where the support vectors of $\mathbf{w}_{\ell_2}$ are associated with positive dual variables $\nu_n$ [28]. Let

$$\kappa(t) = \bar{\mathbf{w}}(t) - \log\left(\frac{2}{N}D^2\alpha^{2D-2}t\right)\mathbf{w}_{\ell_2} - \tilde{\mathbf{w}}. \tag{40}$$

Then

$$\frac{d\kappa(t)}{dt} = \frac{d\bar{\mathbf{w}}(t)}{dt} - \frac{1}{t}\mathbf{w}_{\ell_2}$$

and thus

$$\frac{1}{2}\frac{d}{dt}\|\kappa(t)\|_2^2 = \left(\frac{d\kappa(t)}{dt}\right)^\top\kappa(t)$$

$$= \left(\frac{d\bar{\mathbf{w}}(t)}{dt} - \frac{1}{t}\mathbf{w}_{\ell_2}\right)^\top\kappa(t)$$

$$\overset{(34)}{=} \frac{2}{N}D^2\alpha^{2D-2}\sum_{n=1}^{N}\exp\left(-\mathbf{x}_n^\top\bar{\mathbf{w}}(t)\right)\mathbf{x}_n^\top\kappa(t) - \frac{1}{t}\mathbf{w}_{\ell_2}^\top\kappa(t)$$

$$= \left[\frac{2}{N}D^2\alpha^{2D-2}\sum_{n \in S_2}\exp\left(-\mathbf{x}_n^\top\bar{\mathbf{w}}(t)\right)\mathbf{x}_n^\top\kappa(t) - \frac{1}{t}\mathbf{w}_{\ell_2}^\top\kappa(t)\right]$$

$$+ \left[\frac{2}{N}D^2\alpha^{2D-2}\sum_{n \notin S_2}\exp\left(-\mathbf{x}_n^\top\bar{\mathbf{w}}(t)\right)\mathbf{x}_n^\top\kappa(t)\right] \tag{41}$$

For $n \in S_2$ we have that $\mathbf{x}_n^\top\mathbf{w}_{\ell_2} = 1$, thus

$$\exp\left(-\mathbf{x}_n^\top\bar{\mathbf{w}}(t)\right) = \exp\left(-\mathbf{x}_n^\top\left(\log\left(\frac{2}{N}D^2\alpha^{2D-2}t\right)\mathbf{w}_{\ell_2} + \tilde{\mathbf{w}} + \kappa(t)\right)\right)$$

$$= \frac{1}{\frac{2}{N}D^2\alpha^{2D-2}t}\exp\left(-\mathbf{x}_n^\top\tilde{\mathbf{w}}\right)\exp\left(-\mathbf{x}_n^\top\kappa(t)\right)$$

$$= \frac{1}{\frac{2}{N}D^2\alpha^{2D-2}t}\nu_n\exp\left(-\mathbf{x}_n^\top\kappa(t)\right)$$

and the first bracketed term in eq. (41) can be written as

$$\frac{2}{N}D^2\alpha^{2D-2}\sum_{n\in S_2}\frac{1}{\frac{2}{N}D^2\alpha^{2D-2}t}\nu_n\exp\left(-\mathbf{x}_n^\top\kappa\left(t\right)\right)\mathbf{x}_n^\top\kappa\left(t\right)-\frac{1}{t}\sum_{n\in S_2}\nu_n\mathbf{x}_n^\top\kappa\left(t\right)$$

$$=\frac{1}{t}\sum_{n\in S_2}\left[\nu_n\left(\exp\left(-\mathbf{x}_n^\top\kappa\left(t\right)\right)-1\right)\mathbf{x}_n^\top\kappa\left(t\right)\right]$$

$$\leq 0 \tag{42}$$

since $(e^{-z}-1)z\leq 0$ for all $z$.

Let $\theta=\min_{n\notin S_2}\left(\mathbf{x}_n^\top\mathbf{w}_{\ell_2}\right)>1$ and $c_1=\max_{n\notin S_2}\exp\left(-\mathbf{x}_n^\top\tilde{\mathbf{w}}\right)$. For $n\notin S_2$ we have that

$$\exp\left(-\mathbf{x}_n^\top\bar{\mathbf{w}}\left(t\right)\right)=\exp\left(-\mathbf{x}_n^\top\left(\log\left(\frac{2}{N}D^2\alpha^{2D-2}t\right)\mathbf{w}_{\ell_2}+\tilde{\mathbf{w}}+\kappa\left(t\right)\right)\right)$$

$$\leq\frac{1}{\left(\frac{2}{N}D^2\alpha^{2D-2}t\right)^\theta}\exp\left(-\mathbf{x}_n^\top\tilde{\mathbf{w}}\right)\exp\left(-\mathbf{x}_n^\top\kappa\left(t\right)\right)$$

$$\leq\frac{c_1}{\left(\frac{2}{N}D^2\alpha^{2D-2}t\right)^\theta}\exp\left(-\mathbf{x}_n^\top\kappa\left(t\right)\right)$$

and thus the second bracketed term in eq. (41) can be bounded as following

$$\frac{2}{N}D^2\alpha^{2D-2}\sum_{n\notin S_2}\exp\left(-\mathbf{x}_n^\top\bar{\mathbf{w}}\left(t\right)\right)\mathbf{x}_n^\top\kappa\left(t\right)$$

$$\leq\frac{\frac{2}{N}D^2\alpha^{2D-2}c_1}{\left(\frac{2}{N}D^2\alpha^{2D-2}t\right)^\theta}\sum_{n\notin S_2}\exp\left(-\mathbf{x}_n^\top\kappa\left(t\right)\right)\mathbf{x}_n^\top\kappa\left(t\right)$$

$$\leq\frac{2D^2\alpha^{2D-2}c_1}{\left(\frac{2}{N}D^2\alpha^{2D-2}t\right)^\theta} \tag{43}$$

since $e^{-z}z\leq 1$ for all $z$. Substituting eqs. (42) and (43) in eq. (41) we get

$$\frac{1}{2}\frac{d}{dt}\left\|\kappa\left(t\right)\right\|_2^2\leq\frac{2D^2\alpha^{2D-2}c_1}{\left(\frac{2}{N}D^2\alpha^{2D-2}t\right)^\theta}.$$

Using (39) we get

$$\frac{1}{2}\frac{d}{dt}\left\|\kappa\left(t\right)\right\|_2^2\leq\frac{2D^2\alpha^{2D-2}c_1}{\left(\frac{2}{N}D^2\alpha^{2D-2}\frac{\exp(\bar{\bar{\gamma}}(t))-1}{2D^2\alpha^{2D-2}x_{\max}^2}\right)^\theta}=\frac{2D^2\alpha^{2D-2}c_1}{\left(\frac{1}{N}\frac{\exp(\bar{\bar{\gamma}}(t))-1}{x_{\max}^2}\right)^\theta}.$$

We change variables $t\to\bar{\bar{\gamma}}$ and get

$$\frac{1}{2}\frac{d}{d\bar{\bar{\gamma}}}\left\|\kappa\left(\bar{\bar{\gamma}}\right)\right\|_2^2=\frac{1}{2}\frac{d}{dt}\left\|\kappa\left(\bar{\bar{\gamma}}\left(t\right)\right)\right\|_2^2\frac{dt}{d\bar{\bar{\gamma}}}$$

$$\overset{(37)}{\leq}\frac{2D^2\alpha^{2D-2}c_1}{\left(\frac{1}{N}\frac{\exp(\bar{\bar{\gamma}})-1}{x_{\max}^2}\right)^\theta}\frac{1}{2D^2\alpha^{2D-2}\gamma_2^2\exp\left(-\bar{\bar{\gamma}}\right)}$$

$$=\frac{c_1\exp\left(-\left(\theta-1\right)\bar{\bar{\gamma}}\right)}{\gamma_2^2\left(\frac{1}{N}\frac{1-\exp(-\bar{\bar{\gamma}})}{x_{\max}^2}\right)^\theta}$$

$$\leq C\frac{\exp\left(-\left(\theta-1\right)\bar{\bar{\gamma}}\right)}{\left(1-\exp\left(-\bar{\bar{\gamma}}\right)\right)^\theta}$$

where $C$ is a constant. Integrating we have that for all $\bar{\bar{\gamma}}_0 > 0, \bar{\bar{\gamma}} > \bar{\bar{\gamma}}_0$

$$\|\kappa(\bar{\bar{\gamma}})\|_2^2 - \|\kappa(\bar{\bar{\gamma}}_0)\|_2^2 \leq C \int_{\bar{\bar{\gamma}}_0}^{\bar{\bar{\gamma}}} \frac{\exp\left(-(\theta-1)\bar{\bar{\gamma}}_1\right)}{(1-\exp(-\bar{\bar{\gamma}}_1))^\theta} d\bar{\bar{\gamma}}_1$$

$$\leq C \int_{\bar{\bar{\gamma}}_0}^{\infty} \frac{\exp\left(-(\theta-1)\bar{\bar{\gamma}}_1\right)}{(1-\exp(-\bar{\bar{\gamma}}_1))^\theta} d\bar{\bar{\gamma}}_1$$

$$= C \frac{1}{(\theta-1)(\exp(\bar{\bar{\gamma}}_0)-1)^{\theta-1}}$$

and thus

$$\|\kappa(\bar{\bar{\gamma}})\|_2 \leq C' \tag{44}$$

where $C'$ is a constant. Finally, for $k \notin S_2$ we have that $\mathbf{x}_k^\top \mathbf{w}_{\ell_2} \geq \theta > 1$ and thus

$$\frac{\mathbf{x}_k^\top \bar{\mathbf{w}}(\bar{\bar{\gamma}})}{\bar{\bar{\gamma}}} \overset{(40)}{=} \frac{\mathbf{x}_k^\top \left(\log\left(\frac{2}{N}D^2\alpha^{2D-2}t\right)\mathbf{w}_{\ell_2} + \tilde{\mathbf{w}} + \kappa(\bar{\bar{\gamma}})\right)}{\bar{\bar{\gamma}}}$$

$$\overset{(39)}{\geq} \frac{\log\left(\frac{2}{N}D^2\alpha^{2D-2}\frac{\exp(\bar{\bar{\gamma}})-1}{2D^2\alpha^{2D-2}x_{\max}^2}\right)\mathbf{x}_k^\top\mathbf{w}_{\ell_2} + \mathbf{x}_k^\top\tilde{\mathbf{w}} + \mathbf{x}_k^\top\kappa(\bar{\bar{\gamma}})}{\bar{\bar{\gamma}}}$$

$$\overset{(44)}{\geq} \frac{\log\left(\frac{\exp(\bar{\bar{\gamma}})-1}{Nx_{\max}^2}\right)\theta - \log c_1 - x_{\max}C'}{\bar{\bar{\gamma}}}$$

$$= \frac{\log\left(\frac{\exp(\bar{\bar{\gamma}})-1}{Nx_{\max}^2}\right)}{\bar{\bar{\gamma}}}\theta - \frac{\log c_1 + x_{\max}C'}{\bar{\bar{\gamma}}}\,.$$

Note that $\dfrac{\log\left(\frac{\exp(\bar{\bar{\gamma}})-1}{Nx_{\max}^2}\right)}{\bar{\bar{\gamma}}}$ is monotonically increasing and

$$\frac{\log\left(\frac{\exp(\bar{\bar{\gamma}})-1}{Nx_{\max}^2}\right)}{\bar{\bar{\gamma}}} \overset{\bar{\bar{\gamma}}\to\infty}{\to} 1\,.$$

Therefore there exists $\bar{\bar{\gamma}}^\star$ (independent of $\alpha$!) such that for $\tilde{\gamma} \geq \bar{\bar{\gamma}}^\star$

$$\frac{\log\left(\frac{\exp(\bar{\bar{\gamma}})-1}{Nx_{\max}^2}\right)}{\bar{\bar{\gamma}}} \geq \frac{3\theta+1}{4\theta}$$

and

$$\frac{\log c_1 + x_{\max}C'}{\bar{\bar{\gamma}}} \leq \frac{\theta-1}{4}\,.$$

It follows that for $\tilde{\gamma} \geq \bar{\bar{\gamma}}^\star$

$$\frac{\mathbf{x}_k^\top \bar{\mathbf{w}}(\bar{\bar{\gamma}})}{\bar{\bar{\gamma}}} \geq \frac{3\theta+1}{4\theta}\theta - \frac{\theta-1}{4} = \frac{\theta+1}{2} \doteq \rho_0 > 1\,.$$

From $\bar{\bar{\gamma}} \geq \bar{\gamma}$ (where $\bar{\gamma}(t) = \min_n\left(\mathbf{x}_n^\top \bar{\mathbf{w}}(t)\right)$) we get

$$\frac{\mathbf{x}_k^\top \bar{\mathbf{w}}(\bar{\bar{\gamma}})}{\bar{\gamma}} \geq \frac{\mathbf{x}_k^\top \bar{\mathbf{w}}(\bar{\bar{\gamma}})}{\bar{\bar{\gamma}}} \geq \rho_0 > 1$$

for $\tilde{\gamma} \geq \bar{\bar{\gamma}}^\star = o\left(\alpha^D\right)$ since $\bar{\bar{\gamma}}^\star$ is independent of $\alpha$.

# F Proofs for $D = 2$

## F.1 Kernel Regime Proof

**Theorem 14** (Theorem 4 for $D = 2$). *For $D = 2$, if $\tilde{\gamma}(\alpha) = o(\alpha^2)$ then*

$$\hat{\mathbf{w}} = \underset{\mathbf{w}}{\operatorname{argmin}} \|\mathbf{w}\|_2 \quad \text{s.t.} \ \forall n : \mathbf{x}_n^\top \mathbf{w} \geq 1 \,.$$

*Proof.* We show convergence of the $\ell_2$ margin $\gamma_2(T_\alpha) = \frac{\min_n (\mathbf{x}_n^\top \mathbf{w}(T_\alpha))}{\|\mathbf{w}(T_\alpha)\|_2}$ to the max-margin $\gamma_2$ when $\alpha \to \infty$. Note that by definition $\gamma_2(T_\alpha) \leq \gamma_2$. Next we show that $\gamma_2(T_\alpha) \geq \gamma_2$ when $\alpha \to \infty$. From eq. (10) we have that

$$\gamma_2(t) = \frac{\min_n (\mathbf{x}_n^\top \mathbf{w}(t))}{\|\mathbf{w}(t)\|_2} \geq \frac{\tilde{\gamma}(t) - \log(N)}{\|\mathbf{w}(t)\|_2} \,. \tag{45}$$

In order to lower bound $\gamma_2(t)$ we derive a lower bound on $\tilde{\gamma}(t)$ and an upper bound on $\|\mathbf{w}(t)\|_2$.

**Lower bound on $\tilde{\gamma}(t)$:** Combining (18) and (19) we get

$$\frac{d\tilde{\gamma}(t)}{dt} \geq 8\alpha^2 \gamma_2 \|X\mathbf{r}(t)\|_2$$

$$\Rightarrow \tilde{\gamma}(t) \geq 8\alpha^2 \gamma_2 \int_0^t \|X\mathbf{r}(\tau)\|_2 \, d\tau \,. \tag{46}$$

**Upper bound on $\|\mathbf{w}(t)\|_2$:** We decompose $\dot{\mathbf{w}}(t)$ to two terms:

$$\dot{\mathbf{w}}(t) = 4\sqrt{\mathbf{w}^2(t) + 4\alpha^4 \mathbf{1}} \circ X\mathbf{r}(t)$$

$$= 4\left(\sqrt{\mathbf{w}^2(t) + 4\alpha^4 \mathbf{1}} - 2\alpha^2 \mathbf{1}\right) \circ X\mathbf{r}(t) + 8\alpha^2 X\mathbf{r}(t)$$

$$\Rightarrow \|\dot{\mathbf{w}}(t)\|_2 \leq 4 \left\|\left(\sqrt{\mathbf{w}^2(t) + 4\alpha^4 \mathbf{1}} - 2\alpha^2 \mathbf{1}\right) \circ X\mathbf{r}(t)\right\|_2 + 8\alpha^2 \|X\mathbf{r}(t)\|_2$$

$$\Rightarrow \|\mathbf{w}(t)\|_2 \leq 4 \int_0^t \left\|\left(\sqrt{\mathbf{w}^2(\tau) + 4\alpha^4 \mathbf{1}} - 2\alpha^2 \mathbf{1}\right) \circ X\mathbf{r}(\tau)\right\|_2 d\tau + 8\alpha^2 \int_0^t \|X\mathbf{r}(\tau)\|_2 \, d\tau \,. \tag{47}$$

Let $v(t) = \left\|\left(\sqrt{\mathbf{w}^2(t) + 4\alpha^4 \mathbf{1}} - 2\alpha^2 \mathbf{1}\right) \circ X\mathbf{r}(t)\right\|_2$. Then

$$v(t) \leq \left\|\sqrt{\mathbf{w}^2(t) + 4\alpha^4 \mathbf{1}} - 2\alpha^2 \mathbf{1}\right\|_\infty x_{\max} \|\mathbf{r}(t)\|_1$$

$$= \left\|\sqrt{\mathbf{w}^2(t) + 4\alpha^4 \mathbf{1}} - 2\alpha^2 \mathbf{1}\right\|_\infty x_{\max} \exp(-\tilde{\gamma}(t)) \,.$$

Using Lemma 11 we get

$$v(t) \leq \left(2\alpha^2 \sqrt{\sinh^2\left(\frac{\bar{x}}{2\alpha^2 \gamma_2^2} \tilde{\gamma}(t)\right) + 1} - 2\alpha^2\right) x_{\max} \exp(-\tilde{\gamma}(t))$$

$$= 2\alpha^2 \left[\cosh\left(\frac{\bar{x}}{2\alpha^2 \gamma_2^2} \tilde{\gamma}(t)\right) - 1\right] x_{\max} \exp(-\tilde{\gamma}(t)) \,.$$

We are interested in bounding $\int_0^t v(\tau) \, d\tau$. We change variables $t \to \tilde{\gamma}(t)$ and proceed using (20),

$$\int_0^t v(\tau) \, d\tau \leq \int_0^{\tilde{\gamma}(t)} 2\alpha^2 \left[\cosh\left(\frac{\bar{x}}{2\alpha^2 \gamma_2^2} \tilde{\gamma}\right) - 1\right] x_{\max} \exp(-\tilde{\gamma}) \frac{1}{8\alpha^2 \gamma_2^2 \exp(-\tilde{\gamma})} d\tilde{\gamma}$$

$$= \frac{x_{\max}}{4\gamma_2^2} \int_0^{\tilde{\gamma}(t)} \left[\cosh\left(\frac{\bar{x}}{2\alpha^2 \gamma_2^2} \tilde{\gamma}\right) - 1\right] d\tilde{\gamma}$$

$$= \frac{x_{\max}}{4\gamma_2^2} \left[\frac{2\alpha^2 \gamma_2^2}{\bar{x}} \sinh\left(\frac{\bar{x}}{2\alpha^2 \gamma_2^2} \tilde{\gamma}(t)\right) - \tilde{\gamma}(t)\right] \,. \tag{48}$$

Plugging eqs. (48) in (47) we get

$$\|\mathbf{w}(t)\|_2 \leq \frac{x_{\max}}{\gamma_2^2} \left[\frac{2\alpha^2 \gamma_2^2}{\bar{x}} \sinh\left(\frac{\bar{x}}{2\alpha^2 \gamma_2^2} \tilde{\gamma}(t)\right) - \tilde{\gamma}(t)\right] + 8\alpha^2 \int_0^t \|X\mathbf{r}(\tau)\|_2 \, d\tau \,. \tag{49}$$

**Putting things together:** From eqs. (45) and (12) we have

$$\gamma_2(t) \geq \frac{\tilde{\gamma}(t) - \log(N)}{\|\mathbf{w}(t)\|_2} \geq \frac{\tilde{\gamma}(t)}{\|\mathbf{w}(t)\|_2} - \frac{\log(N)x_{\max}}{\tilde{\gamma}(t)}. \tag{50}$$

Next we set $t = T_\alpha$ and take the limit $\alpha \to \infty$. Note that $\tilde{\gamma}(T_\alpha) \overset{\alpha \to \infty}{\to} \infty$ since $\epsilon(T_\alpha) \overset{\alpha \to \infty}{\to} 0$, and thus the right term in eq. (50) is vanishing. Using eq. (49) we get

$$\lim_{\alpha \to \infty} \frac{1}{\gamma_2(T_\alpha)} \leq \lim_{\alpha \to \infty} \frac{\|\mathbf{w}(T_\alpha)\|_2}{\tilde{\gamma}(T_\alpha)}$$

$$\leq \lim_{\alpha \to \infty} \left[ \frac{x_{\max}}{\gamma_2^2} \left[ \frac{2\alpha^2 \gamma_2^2}{\bar{x}\tilde{\gamma}(T_\alpha)} \sinh\left( \frac{\bar{x}}{2\alpha^2 \gamma_2^2} \tilde{\gamma}(T_\alpha) \right) - 1 \right] + \frac{8\alpha^2}{\tilde{\gamma}(T_\alpha)} \int_0^{T_\alpha} \|X\mathbf{r}(\tau)\|_2 \, d\tau \right].$$

We use $\frac{\tilde{\gamma}(T_\alpha)}{\alpha^2} \overset{\alpha \to \infty}{\to} 0$, $\lim_{z \to 0} \frac{\sinh z}{z} = 1$ and eq. (46) to get

$$\lim_{\alpha \to \infty} \frac{1}{\gamma_2(T_\alpha)} \leq \frac{1}{\gamma_2}.$$

It follows that $\lim_{\alpha \to \infty} \gamma_2(T_\alpha) = \gamma_2$. $\qquad\qquad\qquad\qquad\qquad\qquad\qquad\qquad\qquad\qquad \square$

## F.2 Intermediate Regime Proof

**Theorem 15** (Theorem 6 for $D = 2$). *Under Condition 8, for $D = 2$ if $\lim_{\alpha \to \infty} \frac{\alpha^2}{\tilde{\gamma}(\alpha)} = \mu > 0$, then*

$$\hat{\mathbf{w}} = \underset{\mathbf{w}}{\arg\min}\, Q_\mu^2(\mathbf{w}) \text{ s.t. } \forall n: \mathbf{x}_n^\top \mathbf{w} \geq 1$$

*where $Q_\mu^2(\mathbf{w}) = \sum_{i=1}^d q_2\left(\frac{w_i}{\mu}\right)$ and $q_2(s) = 2 - \sqrt{4 + s^2} + s \cdot \text{arcsinh}\left(\frac{s}{2}\right)$.*

*Proof.* We show that the KKT conditions hold in the limit $\alpha \to \infty$. The KKT conditions are that there exists $\boldsymbol{\nu} \in \mathbb{R}_{\geq 0}^N$ such that

$$\nabla Q_\mu^2(\hat{\mathbf{w}}) = X\boldsymbol{\nu} \tag{51}$$

$$\forall n: \mathbf{x}_n^\top \hat{\mathbf{w}} \geq 1 \tag{52}$$

$$\forall n: \nu_n\left(\mathbf{x}_n^\top \hat{\mathbf{w}} - 1\right) = 0. \tag{53}$$

**Primal feasibility** (52): The condition (52) follows by definition of $\hat{\mathbf{w}}$,

$$\forall n: \mathbf{x}_n^\top \hat{\mathbf{w}} = \lim_{\alpha \to \infty} \frac{\mathbf{x}_n^\top \mathbf{w}(T_\alpha)}{\gamma(T_\alpha)} \geq \lim_{\alpha \to \infty} \frac{\min_n \left(\mathbf{x}_n^\top \mathbf{w}(T_\alpha)\right)}{\gamma(T_\alpha)} = 1. \tag{54}$$

**Stationarity condition** (51): To show the condition (51) let

$$\boldsymbol{\nu} = \frac{4}{\mu} \limsup_{\alpha \to \infty} \int_0^{T_\alpha} \mathbf{r}(s)ds \in \mathbb{R}_{\geq 0}^N. \tag{55}$$

We need to show that

$$\nabla Q_\mu(\hat{\mathbf{w}}) = \frac{1}{\mu} \text{arcsinh}\left(\frac{\hat{\mathbf{w}}}{2\mu}\right) = X\boldsymbol{\nu}.$$

Indeed from eqs. (13) and (11) we have

$$\hat{\mathbf{w}} = \lim_{\alpha \to \infty} \frac{2\alpha^2 \sinh\left(4X \int_0^{T_\alpha} \mathbf{r}(s)ds\right)}{\gamma(T_\alpha)}$$

$$= 2 \lim_{\alpha \to \infty} \frac{\tilde{\gamma}(T_\alpha)}{\gamma(T_\alpha)} \lim_{\alpha \to \infty} \frac{\alpha^2}{\tilde{\gamma}(T_\alpha)} \limsup_{\alpha \to \infty} \sinh\left(4X \int_0^{T_\alpha} \mathbf{r}(s)ds\right)$$

$$= 2\mu \sinh\left[\mu X\left(\frac{4}{\mu} \limsup_{\alpha \to \infty} \int_0^{T_\alpha} \mathbf{r}(s)ds\right)\right]$$

$$= 2\mu \sinh(\mu X\boldsymbol{\nu})$$

and thus $\frac{1}{\mu} \text{arcsinh}\left(\frac{\hat{\mathbf{w}}}{2\mu}\right) = X\boldsymbol{\nu}$, as desired.

**Complementary slackness (53):** To show the condition (53) let $k \in [N]$ such that

$$\mathbf{x}_k^\top \hat{\mathbf{w}} > 1 . \tag{56}$$

We need to show that $\nu_k = 0$. We change variables $t \to \tilde{\gamma}(t)$ and using eq. (20) we get

$$\int_0^{T_\alpha} \exp\left(-\mathbf{x}_k^\top \mathbf{w}(s)\right) ds \leq \frac{1}{8\alpha^2\gamma_2^2} \int_0^{\tilde{\gamma}(T_\alpha)} \exp\left(-\mathbf{x}_k^\top \mathbf{w}(\tilde{\gamma}) + \tilde{\gamma}\right) d\tilde{\gamma} . \tag{57}$$

From Condition 8 we know that there exists $\tilde{\gamma}^\star(\alpha) = o\left(\alpha^2\right)$ and $\rho_0 > 1$ such that for large enough $\alpha$ and $\tilde{\gamma} \in [\tilde{\gamma}^\star(\alpha), \tilde{\gamma}(\alpha)]$, $\frac{\mathbf{x}_k^\top \mathbf{w}(\tilde{\gamma})}{\gamma(\tilde{\gamma})} \geq \rho_0$. Let

$$\tilde{\gamma}_1^\star(\alpha) = \max\left(\frac{2\rho_0 \log N}{\rho_0 - 1}, \tilde{\gamma}^\star(\alpha)\right) = o\left(\alpha^2\right)$$

and $\rho_1 = \frac{\rho_0 + 1}{2} > 1$. Then for large enough $\alpha$ and $\tilde{\gamma} \in [\tilde{\gamma}_1^\star(\alpha), \tilde{\gamma}(\alpha)]$, using $\tilde{\gamma} \leq \gamma + \log N$ we get

$$\begin{aligned}
\frac{\mathbf{x}_k^\top \mathbf{w}(\tilde{\gamma})}{\tilde{\gamma}} &= \frac{\mathbf{x}_k^\top \mathbf{w}(\tilde{\gamma})}{\gamma} \frac{\gamma}{\tilde{\gamma}} \\
&\geq \rho_0 \frac{\tilde{\gamma} - \log N}{\tilde{\gamma}} \\
&= \rho_0 - \rho_0 \frac{\log N}{\tilde{\gamma}} \\
&\geq \rho_0 - \rho_0 \frac{\log N}{\frac{2\rho_0 \log N}{\rho_0 - 1}} \\
&= \rho_1 .
\end{aligned}$$

Next we decompose the RHS of eq. (57) as following

$$\begin{aligned}
\frac{1}{8\alpha^2\gamma_2^2} \int_0^{\tilde{\gamma}(T_\alpha)} \exp\left(-\mathbf{x}_k^\top \mathbf{w}(\tilde{\gamma}) + \tilde{\gamma}\right) d\tilde{\gamma} =& \frac{1}{8\alpha^2\gamma_2^2} \int_0^{\tilde{\gamma}_1^\star(\alpha)} \exp\left(-\mathbf{x}_k^\top \mathbf{w}(\tilde{\gamma}) + \tilde{\gamma}\right) d\tilde{\gamma} \\
&+ \frac{1}{8\alpha^2\gamma_2^2} \int_{\tilde{\gamma}_1^\star(\alpha)}^{\tilde{\gamma}(T_\alpha)} \exp\left(-\mathbf{x}_k^\top \mathbf{w}(\tilde{\gamma}) + \tilde{\gamma}\right) d\tilde{\gamma} \\
=& (I) + (II) \tag{58}
\end{aligned}$$

From eq. (10) we have that $\exp\left(-\mathbf{x}_k^\top \mathbf{w}(\tilde{\gamma}) + \tilde{\gamma}\right) \leq N$ and thus

$$(I) \leq \frac{1}{8\alpha^2\gamma_2^2} \int_0^{\tilde{\gamma}_1^\star(\alpha)} N d\tilde{\gamma} = \frac{N\tilde{\gamma}_1^\star(\alpha)}{8\alpha^2\gamma_2^2} \xrightarrow{\alpha \to \infty} 0 \tag{59}$$

since $\tilde{\gamma}_1^\star(\alpha) = o(\alpha^2)$. For the second term in eq. (58) we have for large enough $\alpha$,

$$\begin{aligned}
(II) &= \frac{1}{8\alpha^2\gamma_2^2} \int_{\tilde{\gamma}_1^\star(\alpha)}^{\tilde{\gamma}(T_\alpha)} \exp\left[-\left(\mathbf{x}_k^\top \frac{\mathbf{w}(\tilde{\gamma})}{\tilde{\gamma}} - 1\right)\tilde{\gamma}\right] d\tilde{\gamma} \\
&\leq \frac{1}{8\alpha^2\gamma_2^2} \int_{\tilde{\gamma}_1^\star(\alpha)}^{\tilde{\gamma}(T_\alpha)} \exp\left[-(\rho_1 - 1)\tilde{\gamma}\right] d\tilde{\gamma} \\
&\leq \frac{1}{8\alpha^2\gamma_2^2} \int_0^\infty \exp\left[-(\rho_1 - 1)\tilde{\gamma}\right] d\tilde{\gamma} \\
&= \frac{1}{8\alpha^2\gamma_2^2(\rho_1 - 1)} \xrightarrow{\alpha \to \infty} 0 . \tag{60}
\end{aligned}$$

By substituting eqs. (59) and (60) in eq. (58) we get that $\nu_k = 0$. $\qquad\square$

## F.3 Rich Regime Proof

**Theorem 16** (Theorem 7 for $D = 2$). *Under Condition 9, for $D = 2$ if $\tilde{\gamma}(\alpha) = \omega(\alpha^2)$ then*

$$\hat{\mathbf{w}} = \underset{\mathbf{w}}{\arg\min} \|\mathbf{w}\|_1 \quad \text{s.t. } \forall n : \mathbf{x}_n^\top \mathbf{w} \geq 1. \tag{61}$$

*Proof.* We show that the KKT conditions for the $\ell_1$ max-margin problem (61) hold in the limit $\alpha \to \infty$. The KKT conditions are that there exists $\boldsymbol{\nu}^{(\ell_1)} \in \mathbb{R}_{\geq 0}^N$ such that

$$X\boldsymbol{\nu}^{(\ell_1)} \in \partial^\circ \|\hat{\mathbf{w}}\|_1 \tag{62}$$

$$\forall n : \mathbf{x}_n^\top \hat{\mathbf{w}} \geq 1 \tag{63}$$

$$\forall n : \nu_n^{(\ell_1)} \left(\mathbf{x}_n^\top \hat{\mathbf{w}} - 1\right) = 0. \tag{64}$$

To this end let

$$\boldsymbol{\nu}^{(\ell_1)} = \limsup_{\alpha \to \infty} \frac{4}{\log \frac{\tilde{\gamma}(T_\alpha)}{\alpha^2}} \int_0^{T_\alpha} \mathbf{r}(s)\, ds \in \mathbb{R}_{\geq 0}^N. \tag{65}$$

The proof for the primal feasibility condition (63) appears in eq. (54).

**Stationarity condition** (62):

$$\hat{\mathbf{w}} = \lim_{\alpha \to \infty} \frac{2\alpha^2 \sinh\left(4X \int_0^{T_\alpha} \mathbf{r}(s)\, ds\right)}{\gamma(T_\alpha)}$$

$$\overset{(11)}{=} \lim_{\alpha \to \infty} \frac{2\alpha^2 \sinh\left(\log \frac{\tilde{\gamma}(T_\alpha)}{\alpha^2} \frac{4X}{\log \frac{\tilde{\gamma}(T_\alpha)}{\alpha^2}} \int_0^{T_\alpha} \mathbf{r}(s)\, ds\right)}{\tilde{\gamma}(T_\alpha)}$$

$$= \lim_{\alpha \to \infty} \frac{2 \sinh\left(\log \left(\frac{\tilde{\gamma}(T_\alpha)}{\alpha^2}\right)^{\frac{4X}{\log \frac{\tilde{\gamma}(T_\alpha)}{\alpha^2}} \int_0^{T_\alpha} \mathbf{r}(s) ds}\right)}{\frac{\tilde{\gamma}(T_\alpha)}{\alpha^2}}$$

$$= \lim_{\alpha \to \infty} \frac{2 \sinh\left(\log (g(\alpha))^{z(\alpha)}\right)}{g(\alpha)}, \tag{66}$$

where we defined

$$g(\alpha) = \frac{\tilde{\gamma}(T_\alpha)}{\alpha^2} \in \mathbb{R}$$

$$z(\alpha) = \frac{4X}{\log \frac{\tilde{\gamma}(T_\alpha)}{\alpha^2}} \int_0^{T_\alpha} \mathbf{r}(s)\, ds \in \mathbb{R}^d.$$

Note that from $\lim_{\alpha \to \infty} \frac{\alpha^2}{\tilde{\gamma}(T_\alpha)} = 0$ we have $\lim_{\alpha \to \infty} g(\alpha) = \infty$ and from (65) we get $\limsup_{\alpha \to \infty} z(\alpha) = X\boldsymbol{\nu}^{(\ell_1)}$. In addition, for some $f > 0$ and $a \in \mathbb{R}$:

$$\frac{2 \sinh(\log f^a)}{f} = \frac{f^a - \frac{1}{f^a}}{f} = f^{a-1} - \frac{1}{f^{a+1}}.$$

Therefore in (66) we have,

$$\hat{\mathbf{w}} = \lim_{\alpha \to \infty} \left(g(\alpha)^{z(\alpha)-1} - \frac{1}{g(\alpha)^{z(\alpha)+1}}\right).$$

Next, it is easy to verify that for all $i = 1, ..., d$:

$$\hat{w}_i > 0 \Rightarrow \limsup_{\alpha \to \infty} z_i(\alpha) = 1$$

$$\hat{w}_i < 0 \Rightarrow \limsup_{\alpha \to \infty} z_i(\alpha) = -1$$

$$\hat{w}_i = 0 \Rightarrow -1 \leq \limsup_{\alpha \to \infty} z_i(\alpha) \leq 1$$

and so $X\boldsymbol{\nu}^{(\ell_1)} \in \partial^\circ \|\hat{\mathbf{w}}\|_1$.

**Complementary slackness** (64): We perform similar steps to the proof of the intermediate regime in Appendix F.2. We change variables $t \to \tilde{\gamma}(t)$ and use the weaker Condition 9, where we replace $\tilde{\gamma}^\star(\alpha)$ with $\tilde{\gamma}_1^\star(\alpha)$ and $\rho_0$ with $\rho_1$ (see the proof of the intermediate regime in Appendix F.2). We get that

$$
\frac{4}{N \log \frac{\tilde{\gamma}(T_\alpha)}{\alpha^2}} \int_0^{T_\alpha} \exp\left(-\mathbf{x}_k^\top \mathbf{w}(s)\right) ds \leq \frac{1}{2N\alpha^2\gamma_2^2 \log \frac{\tilde{\gamma}(T_\alpha)}{\alpha^2}} \int_0^{\tilde{\gamma}(T_\alpha)} \exp\left(-\mathbf{x}_k^\top \mathbf{w}(\tilde{\gamma}) + \tilde{\gamma}\right) d\tilde{\gamma}
$$

$$
\leq \frac{1}{2N\alpha^2\gamma_2^2 \log \frac{\tilde{\gamma}(T_\alpha)}{\alpha^2}} \int_0^{\tilde{\gamma}_1^\star(\alpha)} \exp\left(-\mathbf{x}_k^\top \mathbf{w}(\tilde{\gamma}) + \tilde{\gamma}\right) d\tilde{\gamma}
$$

$$
+ \frac{1}{2N\alpha^2\gamma_2^2 \log \frac{\tilde{\gamma}(T_\alpha)}{\alpha^2}} \int_{\tilde{\gamma}_1^\star(\alpha)}^{\tilde{\gamma}(T_\alpha)} \exp\left(-\mathbf{x}_k^\top \mathbf{w}(\tilde{\gamma}) + \tilde{\gamma}\right) d\tilde{\gamma}
$$

$$
= (I) + (II) \tag{67}
$$

Using $\tilde{\gamma}_1^\star(\alpha) = o\left(\alpha^2 \log \frac{\tilde{\gamma}(T_\alpha)}{\alpha^2}\right)$ we bound the first term similarly to eq. (59):

$$
(I) \leq \frac{\tilde{\gamma}_1^\star(\alpha)}{2N\alpha^2\gamma_2^2 \log \frac{\tilde{\gamma}(T_\alpha)}{\alpha^2}} \xrightarrow{\alpha\to\infty} 0 . \tag{68}
$$

The second term is bounded similarly to eq. (60):

$$
(II) \leq \frac{1}{2N\alpha^2\gamma_2^2 \log \frac{\tilde{\gamma}(T_\alpha)}{\alpha^2} (\rho_1 - 1)} \xrightarrow{\alpha\to\infty} 0 . \tag{69}
$$

By substituting eqs. (68) and (69) in eq. (67) we get that $\nu_k^{(\ell_1)} = 0$. $\qquad\square$

# G   Proofs for $D > 2$

## G.1   Kernel Regime Proof

**Theorem 17** (Theorem 4 for $D > 2$). *For $D > 2$, if $\tilde{\gamma}(\alpha) = o(\alpha^D)$ then*

$$
\hat{\mathbf{w}} = \underset{\mathbf{w}}{\operatorname{argmin}} \|\mathbf{w}\|_2 \quad \text{s.t.} \ \forall n : \ \mathbf{x}_n^\top \mathbf{w} \geq 1 .
$$

The proof is similar in spirit to the proof for the case $D = 2$ (see Appendix F.1).

*Proof.* We show convergence of the $\ell_2$ margin $\gamma_2(T_\alpha) = \frac{\min_n\left(\mathbf{x}_n^\top \mathbf{w}(T_\alpha)\right)}{\|\mathbf{w}(T_\alpha)\|_2}$ to the max-margin $\gamma_2$ as $\alpha \to \infty$. We have that

$$
\gamma_2(t) = \frac{\min_n\left(\mathbf{x}_n^\top \mathbf{w}(t)\right)}{\|\mathbf{w}(t)\|_2} \geq \frac{\tilde{\gamma}(t) - \log(N)}{\|\mathbf{w}(t)\|_2} . \tag{70}
$$

**Lower bound on $\tilde{\gamma}(t)$:** Combining eqs. (28) and (19) we get

$$
\frac{d\tilde{\gamma}(t)}{dt} \geq 2D^2\alpha^{2D-2}\gamma_2 \|X\mathbf{r}(t)\|_2
$$

$$
\Rightarrow \tilde{\gamma}(t) \geq 2D^2\alpha^{2D-2}\gamma_2 \int_0^t \|X\mathbf{r}(\tau)\|_2 d\tau . \tag{71}
$$

**Upper bound on $\|\mathbf{w}(t)\|_2$:** We decompose $\dot{\mathbf{w}}(t)$ to two terms:

$$
\dot{\mathbf{w}}(t) = \alpha^{2D-2}D(D-2)h_D'\left(h_D^{-1}\left(\frac{\mathbf{w}(t)}{\alpha^D}\right)\right) \circ X\mathbf{r}(t)
$$

$$
= \alpha^{2D-2}D^2\left[\frac{D-2}{D}h_D'\left(h_D^{-1}\left(\frac{\mathbf{w}(t)}{\alpha^D}\right)\right) - 2\cdot\mathbf{1}\right] \circ X\mathbf{r}(t) + 2D^2\alpha^{2D-2}X\mathbf{r}(t)
$$

$$\Rightarrow \|\dot{\mathbf{w}}(t)\|_2 \leq \alpha^{2D-2} D^2 \left\| \left[ \frac{D-2}{D} h_D' \left( h_D^{-1} \left( \frac{\mathbf{w}(t)}{\alpha^D} \right) \right) - 2 \cdot \mathbf{1} \right] \circ X\mathbf{r}(t) \right\|_2 + 2D^2 \alpha^{2D-2} \|X\mathbf{r}(t)\|_2$$

$$\Rightarrow \|\mathbf{w}(t)\|_2 \leq \alpha^{2D-2} D^2 \int_0^t \left\| \left[ \frac{D-2}{D} h_D' \left( h_D^{-1} \left( \frac{\mathbf{w}(\tau)}{\alpha^D} \right) \right) - 2 \cdot \mathbf{1} \right] \circ X\mathbf{r}(\tau) \right\|_2 d\tau$$

$$+ 2D^2 \alpha^{2D-2} \int_0^t \|X\mathbf{r}(\tau)\|_2 d\tau . \tag{72}$$

Let $v(t) = \left\| \left[ \frac{D-2}{D} h_D' \left( h_D^{-1} \left( \frac{\mathbf{w}(t)}{\alpha^D} \right) \right) - 2 \cdot \mathbf{1} \right] \circ X\mathbf{r}(t) \right\|_2$. Then

$$v(t) \leq \left\| \frac{D-2}{D} h_D' \left( h_D^{-1} \left( \frac{\mathbf{w}(t)}{\alpha^D} \right) \right) - 2 \cdot \mathbf{1} \right\|_\infty x_{\max} \|\mathbf{r}(t)\|_1$$

$$= \left\| \frac{D-2}{D} h_D' \left( h_D^{-1} \left( \frac{\mathbf{w}(t)}{\alpha^D} \right) \right) - 2 \cdot \mathbf{1} \right\|_\infty x_{\max} \exp(-\tilde{\gamma}(t)) .$$

Using Lemma 13 we get

$$v(t) \leq \left[ \frac{D-2}{D} h_D' \left( \frac{(D-2)\bar{x}}{2D\gamma_2^2 \alpha^D} \tilde{\gamma}(t) \right) - 2 \right] x_{\max} \exp(-\tilde{\gamma}(t)) .$$

We are interested in bounding $\int_0^t v(\tau) d\tau$. We change variables $t \to \tilde{\gamma}(t)$ and proceed using eq. (29),

$$\int_0^t v(\tau) d\tau \leq \int_0^{\tilde{\gamma}(t)} \left[ \frac{D-2}{D} h_D' \left( \frac{(D-2)\bar{x}}{2D\gamma_2^2 \alpha^D} \tilde{\gamma} \right) - 2 \right] \frac{x_{\max} \exp(-\tilde{\gamma})}{2\alpha^{2D-2} D^2 \gamma_2^2 \exp(-\tilde{\gamma})} d\tilde{\gamma}$$

$$= \frac{x_{\max}}{2\alpha^{2D-2} D^2 \gamma_2^2} \int_0^{\tilde{\gamma}(t)} \left[ \frac{D-2}{D} h_D' \left( \frac{(D-2)\bar{x}}{2D\gamma_2^2 \alpha^D} \tilde{\gamma} \right) - 2 \right] d\tilde{\gamma}$$

$$= \frac{x_{\max}}{2\alpha^{2D-2} D^2 \gamma_2^2} \left[ \frac{2\gamma_2^2 \alpha^D}{\bar{x}} h_D \left( \frac{(D-2)\bar{x}}{2D\gamma_2^2 \alpha^D} \tilde{\gamma}(t) \right) - 2\tilde{\gamma}(t) \right] . \tag{73}$$

Plugging eq. (73) in eq. (72) we get

$$\|\mathbf{w}(t)\|_2 \leq \frac{x_{\max}}{2\gamma_2^2} \left[ \frac{2\gamma_2^2 \alpha^D}{\bar{x}} h_D \left( \frac{(D-2)\bar{x}}{2D\gamma_2^2 \alpha^D} \tilde{\gamma}(t) \right) - 2\tilde{\gamma}(t) \right] + 2D^2 \alpha^{2D-2} \int_0^t \|X\mathbf{r}(\tau)\|_2 d\tau . \tag{74}$$

**Putting things together:** From eqs. (70) and (12) we have

$$\gamma_2(t) \geq \frac{\tilde{\gamma}(t) - \log(N)}{\|\mathbf{w}(t)\|_2} \geq \frac{\tilde{\gamma}(t)}{\|\mathbf{w}(t)\|_2} - \frac{\log(N) x_{\max}}{\tilde{\gamma}(t)} . \tag{75}$$

Next we set $t = T_\alpha$ and take the limit $\alpha \to \infty$. Note that $\tilde{\gamma}(T_\alpha) \overset{\alpha \to \infty}{\to} \infty$ since $\epsilon(T_\alpha) \overset{\alpha \to \infty}{\to} 0$, and thus the right term in eq. (75) is vanishing. Using eq. (74) we get

$$\lim_{\alpha \to \infty} \frac{1}{\gamma_2(T_\alpha)} \leq \lim_{\alpha \to \infty} \frac{\|\mathbf{w}(T_\alpha)\|_2}{\tilde{\gamma}(T_\alpha)}$$

$$\leq \lim_{\alpha \to \infty} \left[ \frac{x_{\max}}{2\gamma_2^2} \left[ \frac{2\gamma_2^2 \alpha^D}{\bar{x}\tilde{\gamma}(T_\alpha)} h_D \left( \frac{(D-2)\bar{x}}{2D\gamma_2^2 \alpha^D} \tilde{\gamma}(T_\alpha) \right) - 2 \right] + \frac{2D^2 \alpha^{2D-2}}{\tilde{\gamma}(T_\alpha)} \int_0^{T_\alpha} \|X\mathbf{r}(\tau)\|_2 d\tau \right] .$$

We use $\frac{\tilde{\gamma}(T_\alpha)}{\alpha^2} \overset{\alpha \to \infty}{\to} 0$, eq. (9) and eq. (71) to get

$$\lim_{\alpha \to \infty} \frac{1}{\gamma_2(T_\alpha)} \leq \frac{1}{\gamma_2} .$$

It follows that $\lim_{\alpha \to \infty} \gamma_2(T_\alpha) = \gamma_2$. $\qquad \square$

## G.2 Intermediate Regime Proof

**Theorem 18** (Theorem 6 for $D > 2$). *Under Condition 8, for $D > 2$ if $\lim_{\alpha \to \infty} \frac{\alpha^D}{\tilde{\gamma}(\alpha)} = \mu > 0$, then*

$$\hat{\mathbf{w}} = \arg\min_{\mathbf{w}} Q_\mu^D (\mathbf{w}) \text{ s.t. } \forall n : \mathbf{x}_n^\top \mathbf{w} \geq 1$$

*where $Q_\mu^D (\mathbf{w}) = \sum_{i=1}^d q_D \left( \frac{w_i}{\mu} \right)$ and $q_D (s) = \int_0^s h_D^{-1}(z) \, dz$ for $h_D(z) = (1-z)^{-\frac{D}{D-2}} - (1+z)^{-\frac{D}{D-2}}$.*

*Proof.* The proof is similar in spirit to the proof for the case $D = 2$ (see Appendix F.2). We show that the KKT conditions hold in the limit $\alpha \to \infty$. The KKT conditions are that there exists $\boldsymbol{\nu} \in \mathbb{R}_{\geq 0}^N$ such that

$$\nabla Q_\mu^D (\hat{\mathbf{w}}) = X \boldsymbol{\nu} \tag{76}$$

$$\forall n : \mathbf{x}_n^\top \hat{\mathbf{w}} \geq 1 \tag{77}$$

$$\forall n : \nu_n \left( \mathbf{x}_n^\top \hat{\mathbf{w}} - 1 \right) = 0. \tag{78}$$

The proof of primal feasibility (77) for $D = 2$ applies also here.

**Stationarity condition** (76): Let

$$\boldsymbol{\nu} = \frac{D(D-2)}{\mu} \limsup_{\alpha \to \infty} \left( \alpha^{D-2} \int_0^{T_\alpha} \mathbf{r}(s) \, ds \right) \in \mathbb{R}_{\geq 0}^N. \tag{79}$$

We need to show that

$$\nabla Q_\mu^D (\mathbf{w}) = \frac{1}{\mu} h_D^{-1} \left( \frac{\mathbf{w}}{\mu} \right) = X \boldsymbol{\nu}.$$

Indeed using Lemma 12 and eq. (11) we have

$$\hat{\mathbf{w}} = \lim_{\alpha \to \infty} \frac{\alpha^D h_D \left( \alpha^{D-2} D(D-2) X \int_0^{T_\alpha} \mathbf{r}(s) \, ds \right)}{\gamma(T_\alpha)}$$

$$= \lim_{\alpha \to \infty} \frac{\tilde{\gamma}(T_\alpha)}{\gamma(T_\alpha)} \lim_{\alpha \to \infty} \frac{\alpha^D}{\tilde{\gamma}(T_\alpha)} \limsup_{\alpha \to \infty} h_D \left( \alpha^{D-2} D(D-2) X \int_0^{T_\alpha} \mathbf{r}(s) \, ds \right)$$

$$= \mu h_D \left[ \mu X \left( \frac{D(D-2)}{\mu} \limsup_{\alpha \to \infty} \left( \alpha^{D-2} \int_0^{T_\alpha} \mathbf{r}(s) \, ds \right) \right) \right]$$

$$= \mu h_D (\mu X \boldsymbol{\nu})$$

and thus $\frac{1}{\mu} h_D^{-1} \left( \frac{\hat{\mathbf{w}}}{\mu} \right) = X \boldsymbol{\nu}$, as desired.

**Complementary slackness** (78): Let $k \in [N]$ such that

$$\mathbf{x}_k^\top \hat{\mathbf{w}} > 1. \tag{80}$$

We have to show that $\nu_k = 0$. We change variables $t \to \tilde{\gamma}(t)$ and using eq. (29) we get

$$\alpha^{D-2} \int_0^{T_\alpha} \exp \left( -\mathbf{x}_k^\top \mathbf{w}(s) \right) ds \leq \frac{1}{2D^2 \alpha^D \gamma_2^2} \int_0^{\tilde{\gamma}(T_\alpha)} \exp \left( -\mathbf{x}_k^\top \mathbf{w}(\tilde{\gamma}) + \tilde{\gamma} \right) d\tilde{\gamma}. \tag{81}$$

Next we decompose the RHS of eq. (81) and employ Condition 8, where similarly to the case $D = 2$ we replace $\tilde{\gamma}^\star(\alpha)$ with $\tilde{\gamma}_1^\star(\alpha)$ and $\rho_0$ with $\rho_1$ (see the proof of the intermediate regime for $D = 2$ in Appendix F.2). We get that

$$\frac{1}{2D^2 \alpha^D \gamma_2^2} \int_0^{\tilde{\gamma}(T_\alpha)} \exp \left( -\mathbf{x}_k^\top \mathbf{w}(\tilde{\gamma}) + \tilde{\gamma} \right) d\tilde{\gamma} = \frac{1}{2D^2 \alpha^D \gamma_2^2} \int_0^{\tilde{\gamma}_1^\star(\alpha)} \exp \left( -\mathbf{x}_k^\top \mathbf{w}(\tilde{\gamma}) + \tilde{\gamma} \right) d\tilde{\gamma}$$

$$+ \frac{1}{2D^2 \alpha^D \gamma_2^2} \int_{\tilde{\gamma}_1^\star(\alpha)}^{\tilde{\gamma}(T_\alpha)} \exp \left( -\mathbf{x}_k^\top \mathbf{w}(\tilde{\gamma}) + \tilde{\gamma} \right) d\tilde{\gamma}$$

$$= (I) + (II) \tag{82}$$

From eq. (10) we have that $\exp\left(-\mathbf{x}_k^\top \mathbf{w}\left(\tilde{\gamma}\right) + \tilde{\gamma}\right) \leq N$ and thus

$$(I) \leq \frac{1}{2D^2\alpha^D\gamma_2^2} \int_0^{\tilde{\gamma}_1^\star(\alpha)} N d\tilde{\gamma} = \frac{N\tilde{\gamma}_1^\star(\alpha)}{2D^2\alpha^D\gamma_2^2} \overset{\alpha\to\infty}{\to} 0 \tag{83}$$

since $\tilde{\gamma}_1^\star(\alpha) = o(\alpha^D)$. For the second term in eq. (82) we get for large enough $\alpha$,

$$\begin{aligned}
(II) &= \frac{1}{2D^2\alpha^D\gamma_2^2} \int_{\tilde{\gamma}_1^\star(\alpha)}^{\tilde{\gamma}(T_\alpha)} \exp\left[-\left(\mathbf{x}_k^\top \frac{\mathbf{w}\left(\tilde{\gamma}\right)}{\tilde{\gamma}} - 1\right)\tilde{\gamma}\right] d\tilde{\gamma} \\
&\leq \frac{1}{2D^2\alpha^D\gamma_2^2} \int_{\tilde{\gamma}_1^\star(\alpha)}^{\tilde{\gamma}(T_\alpha)} \exp\left[-\left(\rho_1 - 1\right)\tilde{\gamma}\right] d\tilde{\gamma} \\
&\leq \frac{1}{2D^2\alpha^D\gamma_2^2} \int_0^\infty \exp\left[-\left(\rho_1 - 1\right)\tilde{\gamma}\right] d\tilde{\gamma} \\
&= \frac{1}{2D^2\alpha^D\gamma_2^2\left(\rho_1 - 1\right)} \overset{\alpha\to\infty}{\to} 0 \,.
\end{aligned} \tag{84}$$

By substituting eqs. (83) and (84) in eq. (82) and back in eq. (81) we get that $\nu_k = 0$. $\qquad\square$

### G.3   Rich Regime Proof

**Theorem 19** (Theorem 7 for $D > 2$). *Under Condition 8, for $D > 2$ if $\tilde{\gamma}(\alpha) = \omega(\alpha^D)$ then*

$$\hat{\mathbf{w}} = \underset{\mathbf{w}}{\operatorname{argmin}} \|\mathbf{w}\|_1 \quad \text{s.t. } \forall n : \mathbf{x}_n^\top \mathbf{w} \geq 1 \,. \tag{85}$$

*Proof.* We show that the KKT conditions (62), (63), (64) for the $\ell_1$ max-margin problem (85) hold in the limit $\alpha \to \infty$. To this end let

$$\boldsymbol{\nu}^{(\ell_1)} = D\left(D - 2\right) \limsup_{\alpha\to\infty} \left(\alpha^{D-2} \int_0^{T_\alpha} \mathbf{r}\left(s\right) ds\right) \in \mathbb{R}_{\geq 0}^N \,. \tag{86}$$

Note that this definition is similar to eq. (79), and if $\nu_k = 0$ for some $k$ then also $\nu_k^{(\ell_1)} = 0$. Therefore it is left to show the stationarity condition $X\boldsymbol{\nu}^{(\ell_1)} \in \partial^\circ \|\hat{\mathbf{w}}\|_1$. Indeed, from eq. (24) we know that $-1 \leq \left[X\boldsymbol{\nu}^{(\ell_1)}\right]_i \leq 1$ for all $i$. In addition

$$\begin{aligned}
\hat{\mathbf{w}} &= \lim_{\alpha\to\infty} \frac{\alpha^D h_D\left(\alpha^{D-2}D\left(D-2\right)X\int_0^{T_\alpha}\mathbf{r}\left(s\right)ds\right)}{\gamma\left(T_\alpha\right)} \\
&\overset{(11)}{=} \lim_{\alpha\to\infty} \left(\frac{\alpha^D}{\tilde{\gamma}\left(T_\alpha\right)} h_D\left(\alpha^{D-2}D\left(D-2\right)X\int_0^{T_\alpha}\mathbf{r}\left(s\right)ds\right)\right)\,.
\end{aligned}$$

Assume that $\hat{w}_i > 0$. As $\lim_{\alpha\to\infty}\left(\frac{\alpha^D}{\tilde{\gamma}\left(T_\alpha\right)}\right) = 0$ we must have that

$$\left[\alpha^{D-2}D\left(D-2\right)X\int_0^{T_\alpha}\mathbf{r}\left(s\right)ds\right]_i \overset{\alpha\to\infty}{\to} 1$$

and thus $\left[X\boldsymbol{\nu}^{(\ell_1)}\right]_i \overset{\alpha\to\infty}{\to} 1$. Similarly, if $\hat{w}_i < 0$ we get $\left[X\boldsymbol{\nu}^{(\ell_1)}\right]_i \overset{\alpha\to\infty}{\to} -1$. It follows that $X\boldsymbol{\nu}^{(\ell_1)} \in \partial^\circ \|\hat{\mathbf{w}}\|_1$. $\qquad\square$

## H   Additional Simulation Results and Details

### H.1   Optimization trajectories with $\tilde{\gamma}$ indicators

In Figure 6 we repeat the optimization trajectories from Figure 3, but we add indicators that indicate the value of $\tilde{\gamma}$ along the path. Recall that $\tilde{\gamma}(t) = -\log\epsilon(t)$. For example, a number 10 near some point on the path means that the loss at this point is $\exp(-10)$.

In all three examples we observe that for $\alpha = 100$, where the trajectory first visits the $\ell_2$ predictor, around the $\ell_2$ predictor we have $\tilde{\gamma} = 10^4 = \alpha^2$, as suggested by our theoretical results. In the top figure we also plot the path for $\alpha = 10000$, and again around the $\ell_2$ predictor it holds that $\tilde{\gamma} = 10^8 = \alpha^2$.

In addition we can see that in order to be rather close to $\ell_1$ with large initialization, we need very large $\tilde{\gamma}$, corresponding to extremely small loss $\epsilon$. For example, consider the center plot. For $\alpha = 100$ to be close to $\ell_1$ direction we need $\tilde{\gamma} = 10^8$, or $\epsilon = \exp(-10^8)$ ! However, with small initialization, *e.g.*, $\alpha = 0.001$, $\tilde{\gamma}$ can be as small as $0.1$, or $\epsilon = \exp(-0.1) \approx 0.9$, and we are close to $\ell_1$.

### H.2 Understanding the non-unique $\ell_1$ case

In Figure 3(c) we showed an example of optimization trajectories for data with non-unique $\ell_1$ predictor. We can observe that for different initializations, the selected $\ell_1$ direction, and thus the implicit bias, is different.

It is interesting to understand what are the properties of different $\ell_1$ solutions. To this end, in Figure 7 we plot the optimization trajectory in a different way. Instead of looking at the direction of the predictor (as in Figure 3(c)), we consider the excess $\ell_1$ and $\ell_2$ norms along the path, defined as $\|\mathbf{w}(t)\|_1 / \|\mathbf{w}_{\ell_1}\|_1 - 1$ and $\|\mathbf{w}(t)\|_2 / \|\mathbf{w}_{\ell_2}\|_2 - 1$ where $\mathbf{w}_{\ell_1}$ and $\mathbf{w}_{\ell_2}$ are the $\ell_1$ and $\ell_2$ max-margin (minimum norm) solutions accordingly.

We can observe that for large initialization, where we follow the $Q_\mu$ path, the selected $\ell_1$ predictor has the smallest $\ell_2$ norm. Moreover, for small initialization, the selected $\ell_1$ predictor has the largest $\ell_2$ norm. Thus, in this case, we see an example where the asymptotic (at a long time/small loss) implicit bias is affected by the initialization. This in contrast to previous results for exp-tailed losses (*e.g.*, [13, 16, 19, 20, 28]), where the asymptotic bias was independent of the initialization.

### H.3 Local minima in high dimension

In Figure 8 we consider optimization trajectories for data in dimension 10. In this case we cannot show the direction of the predictor $\frac{\mathbf{w}(t)}{\|\mathbf{w}(t)\|_2}$ on a sphere, as we did for data in dimension 3. Instead, we take the approach similar to Figure 7, where we show the excess margins.

We consider two datasets in dimension 10. The first is a random, yet separable, data composed of 10 points where the coordinates are drawn from $\sim \mathcal{U}(0,1)$. The second dataset is a sparse dataset of 4 points, where the first coordinate is 1 and the other 9 coordinates are random noise $\sim \mathcal{U}(0, 0.5)$. This dataset allows a large separation between the $\ell_2$ max-margin and $\ell_1$ max-margin solutions.

We train depth-3 linear diagonal network and plot the optimization trajectories in $\ell_{2/3}$-$\ell_2$ plane. We observe that for the random data (Figure 8(a)), there are many local minima of the max $\ell_{2/3}$ margin, and depending on initialization we are biased towards different local-minima points. However, with a large initialization we converge to a local point, quite close to the $Q_\mu$ path and $\ell_1$.

For the sparse data (Figure 8(b), and a zoom-in shown in Figure 8(c)) the local minima are quite far away from the paths. Also, in this case, the $\ell_1$ and $\ell_{2/3}$ max-margin solutions are the same, and with a large initialization we converge to them, along the $Q_\mu$ path. This seems to suggest that for certain structure data, like sparse data, we tend to converge to the *global* max $\ell_{2/3}$-margin predictor.

### H.4 Tangent kernel during training

In Figure 5(a) we showed how the excess $\ell_1$-norm depends on $\alpha$ and depth $D$, and measured closeness to the rich limit by excess $\ell_1$-norm. An alternative and complementary approach is to look at the tangent kernel $K_t(\mathbf{x}, \mathbf{x}') = \langle \nabla_{\mathbf{u}} f(\mathbf{u}(t), \mathbf{x}), \nabla_{\mathbf{u}} f(\mathbf{u}(t), \mathbf{x}') \rangle$, which is directly related to closeness to the kernel regime. As discussed in Section 2, the tangent kernel is almost fixed in the kernel regime, yet can change significantly when we exit the kernel regime.

In Figure 9 we show the kernel distance during optimization for the same data and network (depth 2) as in Figure 5(a). The kernel distance is defined as $1 - \text{CosineSimilarity}(K(t), K(0))$ where $K(0) \in \mathbb{R}^{N \times N}$ is the tangent kernel at initialization, $K(t) \in \mathbb{R}^{N \times N}$ is the tangent kernel at time $t$,

and

$$\text{CosineSimilarity}\left(K(t), K(0)\right) = \frac{\langle K(t), K(0)\rangle}{\|K(t)\|_2 \|K(0)\|_2} = \frac{\text{Tr}\left(K(t)K^\top(0)\right)}{\|K(t)\|_2 \|K(0)\|_2}.$$

Here we focus on exiting the kernel regime, rather than closeness to the rich regime. We observe that increasing depth will help to exit the kernel regime (where $distance \approx 0$) earlier, at a larger loss value $\epsilon$. Decreasing the initialization has a similar effect, and this is consistent with Figure 5(a).

### H.5 Addressing Numerical Issues

In our simulations we employ the normalized gradient descent update rule, given by

$$\mathbf{u}(t+1) = \mathbf{u}(t) - \eta \frac{\nabla \mathcal{L}(\mathbf{u}(t))}{\mathcal{L}(\mathbf{u}(t))}$$

where $\mathbf{u} \in \mathbb{R}^{2d}$ is the vector of parameters and

$$\mathcal{L}(\mathbf{u}(t)) = \frac{1}{N}\sum_{n=1}^{N} \exp\left(-\tilde{\mathbf{x}}_n^\top \mathbf{u}^D(t)\right).$$

This algorithm effectively enlarges the learning rate according to the current loss, and for single layer linear models Nacson et al. [21] showed that the loss decreases exponentially faster.

Let $G(\mathbf{u}(t)) = \frac{\nabla \mathcal{L}(\mathbf{u}(t))}{\mathcal{L}(\mathbf{u}(t))}$. During training the loss can become extremely small, *e.g.,* well beyond $10^{-1000}$, and in this case also the gradient is very small. This can cause numerical issues in calculating $G$. In order to have a numerically stable evaluation of $G$, and avoid cases like $0/0$, we take an approach similar to [19]. Specifically, let

$$\bar{\gamma}_n(t) = \tilde{\mathbf{x}}_n^\top \mathbf{u}^D(t) \quad, \qquad \gamma(t) = \min_n \bar{\gamma}_n(t)$$

Then we have that

$$\nabla \mathcal{L}(\mathbf{u}(t)) = -\frac{D}{N}\mathbf{u}^{D-1}(t) \circ \sum_{n=1}^{N} \exp\left(-\bar{\gamma}_n(t)\right)\tilde{\mathbf{x}}_n$$

and

$$G(\mathbf{u}(t)) = -D\mathbf{u}^{D-1}(t) \circ \frac{\sum_{n=1}^{N} \exp\left(-\bar{\gamma}_n(t)\right)\tilde{\mathbf{x}}_n}{\sum_{n=1}^{N} \exp\left(-\bar{\gamma}_n(t)\right)}$$

$$= -D\mathbf{u}^{D-1}(t) \circ \frac{\sum_{n=1}^{N} \exp\left(-\left(\bar{\gamma}_n(t) - \gamma(t)\right)\right)\tilde{\mathbf{x}}_n}{\sum_{n=1}^{N} \exp\left(-\left(\bar{\gamma}_n(t) - \gamma(t)\right)\right)}. \tag{87}$$

We calculate $G$ according to (87). Note that $\max_n \exp\left(-\left(\bar{\gamma}_n(t) - \gamma(t)\right)\right) = 1$ so the denominator is at least 1 and the sum in the numerator will contain at least one support vector $\tilde{\mathbf{x}}_{n:\bar{\gamma}_n(t)=\gamma(t)}$.

It is important to note that we never represent the loss values, but only the parameters $\mathbf{u}$. Thus, as long as $\mathbf{u}$ can be represented by `float64` precision, the simulation can continue, and we get extremely large parameters corresponding to an extremely small loss.

Figure 6: The same optimization trajectories from Figure 3 with $\tilde{\gamma}$ values indications.

Figure 7: Optimization trajectories for data in Figure 3(c) in excess $\ell_1$-norm - excess $\ell_2$-norm plane.

(a) Random data

(b) Sparse data

(c) Zoom-in of Figure 8(b)

Figure 8: Optimization trajectories for data in dimension 10.

Figure 9: The Kernel distance is defined as $1 - \mathrm{CosineSimilarity}\,(K(t), K(0))$ where $K(0)$ is the tangent kernel at initialization and $K(t)$ is the tangent kernel at time $t$.