[Reviews · NeurIPS 2020]

Review 1

Summary and Contributions: The paper studies the optimization trajectory in diagonal linear networks, when the data is linearly separable and exponential loss is used for training. In particular, the authors examine the relationship between the optimization solution and the initialization scale. They establish that in this setting, the trajectory can enter the "Rich Regime" only if the training loss is optimized below a certain threshold. Otherwise, the network will be in the kernel regime. In numerical section, the authors evaluate these thresholds for a few simple datasets. These thresholds end up being infinitesimally small. The authors conclude that more work is needed to understand the effective implicit bias in realistic settings.

Strengths: The paper provides an interesting analysis of a simple model that shows complex phenomena similar to what might be happening in real-life network. Further research in this direction can be very fruitful for the community.

Weaknesses: The paper studies a very limited model, under a very limited setting. In this scenario it is possible that some (or most) of the phenomena observed here are just simply by-products of this specific setting. It would have been nice to include a section to discuss the implications of these results on more general settings.

Correctness: On page (3), L(u) is defined as the average loss. But in various point through out the paper, it seems that the factor (1/N) has been discarded. For example, I believe the calculation of line 146 might be using L(u) = \sum_{n=1}^N \ell(y, y'). Similarly, in line 162, it is stated that \epsilon varies in the interval (0, N]. Should it be the case if the loss is normalized by a factor of N? This might be causing some issues. I believe equations (2), (3), and (4) are missing "\forall n" in their constraints.

Clarity: The paper is well-written for the most part. For a person who is not familiar with the work of Woodworth et al, this paper might be a difficult read as I believe some of the definitions and discussions are absent from this manuscript. For example, in section 3, there is no clear statement regarding the algebraic form of the multi-layer diagonal networks. There are minor typos on lines 125 and 155.

Relation to Prior Work: A substantial portion of the paper is spent on discussing previous work. In particular, the results of Chizat et al and Lyu and Li are discussed.

Reproducibility: Yes

Additional Feedback: In the mainstream results on NTK, overparameterization is key in achieving the kernel behavior. In the results presented in the paper, the role of the number of observations, N, seems to be rather limited. Would you mind commenting on that? The introduction compares and contrasts the results of this paper with the results of Chizat and Bach / Lyu and Li. Are these results proven under the same assumptions and settings? -------------- Comments After Authors' Feedback: Thanks for your comments and clarifications.


Review 2

Summary and Contributions: This paper studies the asymptotic trajectories of gradient flow and their implicit bias for diagonal linear networks, a previously established toy model for deep nets. The authors in particular study the transition between the kernel (NTK) and non-kernel regimes and show the dependence of this transition on initialization scale, besides the training accuracy. Interestingly, much of this behavior happens at impractically small accuracies.

Strengths: The very exciting results\ is that the authors identified a transition boundary for the kernel regime and show transition of implicit bias from l_2 to that of l_{D/2} in the "rich" regime, building on previous work on implicit bias. The authors make very tight statements about the trajectories of the continuum version of gradient descent and verify them experimentally, showing exactly the behavior expected from their results. Perhaps even more importantly, the numerical results in section 5 confirm and give quantitative statements to the previous observation that convergence of gradient descent "in direction" happens over ridiculously long time scales.

Weaknesses: Main weakness of this work is simple - do these interesting results about diagonal linear nets extend in any way to deep nonlinear networks, either fully connected or convolutional? It is far from clear what the practical application of this work is. Nonetheless, this is a step in a good direction in understanding implicit bias of gradient descent. Another weakness is the fact that the results depend on the continuum approximation of gradient descent and it's clear that some of the crucial assumptions do not carry over, like for example the guarantee that the training loss is monotonically decreasing along the gradient trajectory (due to overshooting from finite steps). ---------------------------------------- After author response ---------------------------------------- Thank you for responding to my two concerns - while I understand that they still stand, thank you for pointing to Woodworth et al. 2020 [30] for the first one. Regarding the gradient flow point, I would have liked a response to my worry about the importance of monotonicity, but I appreciate the difficulty of this analysis.

Correctness: Despite attempting to, I have been unable to disprove any of the statements, which together with the numerical experiments gives me reasonable confidence of correctness of the results (though I would be surprised if all the numerical factors are correct).

Clarity: With basically all the proofs delegated to the supplementary material, the main text is very readable, with only some minor corrections necessary - see below.

Relation to Prior Work: Any theorems that are adapted are clearly marked and the discussion throughout the paper makes it clear what is a new result.

Reproducibility: Yes

Additional Feedback: Some comments: - In lines 162 and 164, you claim that \epsilon\in(0,N]. Could you clarify why that's the case? It seems to me that while \epsilon starts at 1, there is obviously no upper bound on the exponential loss (without the separability assumption). - (repeat) do you have any intuition about the discrete gradient descent case? - Your use of \mu on lines 195-211 is confusing, since you've previously used \mu=\alpha^D. This could use clarifying. - I think there's a typo in the \epsilon value in line 285. - I'm very impressed with your clever use of the Broader Impact statement. ---------------------------------------- After author response ---------------------------------------- Thank you for clarifying my confusions here. Overall, my opinion that this is a good paper has not changed.


Review 3

Summary and Contributions: This paper studies the implicit bias of gradient flow for deep linear classification problems. In particular, this paper shows that different initialization scale will lead to different implicit bias. More specifically, this paper considers the transition between the rich regime and kernel regime, and precisely characterizes how the implicit bias varies when choosing different target accuracy epsilon and scaling parameter alpha. ####After reading authors' response. The authors have well addressed my concerns, thus I would like to increase my score to 7.

Strengths: This paper proves the implicit bias of gradient flow for deep linear classification under various ranges of the target accuracy epsilon and scaling parameter alpha. This paper is technically sound, the derived theory is interesting and helpful for further understanding of the role that gradient descent plays in deep learning.

Weaknesses: The network model considered in this paper is rather simple, and it seems that considering D-layer diagonal linear network is no more difficult than 2-layer diagonal linear network. The comparison to the related work [16, 30] has not been clearly stated

Correctness: yes

Clarity: yes

Relation to Prior Work: This part has not been clearly discussed in the paper. It remains unclear how the results in the paper compared with two relevant work [16, 30]

Reproducibility: Yes

Additional Feedback: Overall this is a nice work that proves the implicit bias of gradient flow in a broader range of regimes. My major concern is the novelty and impact of the results derived in this paper given existing works. Firstly, the theoretical results are mostly established based on a quite simple model (diagonal linear networks on separable data), which may be difficult to be generalized to more practical settings. Additionally, based on such a simple model, the effect caused by the depth may not be well characterized (currently it seems that this paper only leverages the fact that a D-layer network is D-homogeneous, which has already be well investigated in [19]). Secondly, as [16] has already shown that a deep linear network with fixed initialization scaling trained by gradient flow will finally converge to the max-margin solution (all weight matrices will be rank-1). How does the result proved in this paper compared to it? Moreover, what’s the difference and relationship between the theoretical results and [30], which shows the transition between the kernel regime and rich regime for nonlinear overparameterized models? The authors should precisely state the difference and relevance to those related work.


Review 4

Summary and Contributions: This paper studies asymptotically how the scale of initialization and the training accuracy determine the solution trajectory of gradient descent enters the 'kernel regime' and 'rich regime' when optimize an exp loss in classification problems.

Strengths: As an followup work to [Woodworth 2019], this work extends the understanding of the transition between rich and kernel regime to the classification problem. It also provides some interesting understanding to the good performance of super deep network.

Weaknesses: 1, It considers the asymptotic setting, but this is fine for this line of research. 2. The theory is limited to linear diagonal network. Whether the conclusion can be generalized to nonlinear network remains unknown. 3.In Theorem 4, Lyu and Li [2020])'s result actually holds for every finite initialization. Why we need to make it go to infinity in equation 4? Moreover, in the main results of this paper, it still needs the initialization to be infinity. A finite initialization results may be more interested here in my opinion. 4. I suggest the author to highlight the technical difficulty of the paper compared to [Woodworth 2019] and [Chizat et al, 2020].

Correctness: The claim and proof looks correct to me.

Clarity: The paper is well written and easy to understand.

Relation to Prior Work: It highlights the difference between this paper and the previous works.

Reproducibility: Yes

Additional Feedback:

[Author Response · NeurIPS 2020]

We thank the reviewers for their reviews and comments. We will incorporate your suggestions in the final version.

We analyze an extremely simple model (as explicitly discussed and motivated, e.g. on lines 59-61 and the broader
impact section), but have reason to believe insights, behaviours and analysis methods will carry over to more complex
and realistic models. The paper Woodworth et al. 2020 [30] already shows how phenomena in this simple model
predicted by theory can be observed also for complex realistic networks. Insights about the extreme training accuracy
needed to reach the limiting behaviour are already helping us and others understand apparent discrepancies regarding
matrix factorization models. In addition, as we comment in lines 36-37, the effect of width is equivalent to that of
initialization scale (more precisely: increasing width corresponds to effectively scaling initialization as sqrt(width) in
matrix factorization / wide diagonal nets, see [30]). Therefore, our analysis is also relevant to more-overparameterized
models, with width$\to \infty$, and the number of weights entering whenever we discuss init scale.

**Normalization of the loss function (Reviewers #1 and #2):** The loss is defined to be the average over all examples.
Thus the interval of $\epsilon$ should be (0,1], this is a typo that does not cause difficulty with analysis and we will fix it.

**For reviewer #1:** - There is a typo in the inequality in line 146, we will fix it. Also, in eq (2),(3),(4) it should be $\forall n$,
we will add it. All other minor typos will be fixed, thank you.
- Note that the algebraic form of the model appears in line 147.
- On the role of the number of observations: We analyze the linearly separable case. Overparameterization is a generic
way of achieving separability, but once the data is separable, the number of observations does not play a direct role.
- Comparison to assumptions/setting of Chizat and Bach 2020 [6] / Lyu and Li 2020 [19]: These papers analyze more
general models, as we discuss in Section 2. Our analytic results are only for the linear diagonal nets - which is a special
case of the models in those papers. For this homogeneous model all the assumptions of previous works hold.

**For reviewer #2:** - The discrepancy between gradient flow and discrete gradient descent is indeed very interesting. We
actually think that for large step-sizes one would exit the kernel regime much faster. Characterizing this is an important
challenge, but seems difficult (both we, and others, have tried, and are still trying to study this). In this paper we focus
on the training accuracy ($\epsilon$), but we certainly are interested also in the effects of other hyperparameters, including, and
perhaps most importantly, the step-size.
- Note that $Q_\mu$ is defined for general $\mu$, and $\mu = \alpha^D$ is only for the square loss setting studied in Woodworth et al. [30]
and takes different values in our Theorem 6. We will make it more clear.
- In line 285 it should be $\epsilon = \exp(-10000)$. We will fix that, thank you.

**For reviewer #3:** - Comparison with Ji and Telgarsky 2019 [16] and Lyu and Li 2020 [19]: References [16] and [19]
analyze the asymptotic max-margin solution only for *infinite* training accuracy (ie infinite training time or zero training
loss) and fixed initialization, i.e. the "rich regime", as discussed in the introduction, and in lines 125-140 in Section 2
(including Theorem 2, and footnote 3 where we discuss how [16] fits in). Section 2 is dedicated to discussing how this
is only one extreme endpoint (the other, corresponding to the "kernel regime", occurs when the training accuracy is
finite and the initialization grows to infinity). This is also visualized in Figure 1 and in the table above the figure. As
discussed explicitly in lines 55-58, the submission fills the gaps between the two extremes and studies the transition
between them. Moreover, [16]'s results are for fully connected linear networks where the rich regime and kernel regime
both lead to $L_2$ max-margin, while for the diagonal network, the rich regime limit ($L_1$ or $L_{2/D}$) is very different from
the kernel regime ($L_2$).
- Comparison with Woodworth et al. 2020 [30]: As discussed in the intro in lines 30-33 and in lines 119-125, [30]
analyzes the transition for *regression with the squared loss*, and does not study training accuracy as a controlling hyper-
parameter. The effect of the training accuracy is more important for classification problems since unlike regression,
with exponential loss, no finite parameter can ever achieve zero training loss. Thus, as discussed in lines 40-45, the
classification setting we study is substantially different, and in order to understand it we need to study the effect of
training accuracy, which was not studied in [30].

**For reviewer #4:** - In eq (4) we take $\alpha$ to infinity only since we want to emphasize the importance of the order of limits
when both $\alpha$ and time go to infinity (in contrast with eq. 2).
- Indeed, our theoretical results are asymptotic and obtained when *both* initialization and accuracy go to infinity at some
relative rates. The case of finite initialization and infinite accuracy is already known in Lyu and Li 2020 [19], Nacson et
al. 2019 [20], and so is the case of infinite initialization and finite accuracy Chizat et al. 2019 [7]. We fill in the gap by
letting both go to infinity while maintaining some finite relationship between them. Non-asymptotic results for finite
initialization and finite accuracy seem very difficult to obtain, but we see in figure 5(b) that the relationship predicted by
our asymptotic results is already seen also for moderate finite $\alpha$.
- The technical difficulty compared to [Woodworth et al. 2020] stems from the difference in KKT conditions. [Chizat et
al 2019] derived only the kernel regime (large initialization and finite time).

[Meta-Review · NeurIPS 2020]

All reviewers have found the work well written, technically sound and providing very interesting insights on a simple model that may lead to a better understanding of more complicated models. In particular the connection between the optimization path and the regularization path is insightful, and the numerical experiments nicely illustrate the theory. I thus recommend accept.